# WHEN DO GFLOWNETS
# LEARN THE RIGHT DISTRIBUTION?

**Tiago da Silva**    **Rodrigo Barreto Alves**    **Eliezer de Souza da Silva**
Getulio Vargas Foundation
`{tiago.henrique, rodrigo.alves, eliezer.silva}@fgv.br`

**Amauri Souza**
Federal Institute of Ceará
`amauriholanda@ifce.edu.br`

**Vikas Garg**
YaiYai Ltd and Aalto University
`vgarg@csail.mit.edu`

**Samuel Kaski**
Aalto University, Manchester University
`samuel.kaski@aalto.fi`

**Diego Mesquita**
Getulio Vargas Foundation
`diego.mesquita@fgv.br`

## ABSTRACT

Generative Flow Networks (GFlowNets) are an emerging class of sampling methods for distributions over discrete and compositional objects, e.g., graphs. In spite of their remarkable success in problems such as drug discovery and phylogenetic inference, the question of when and whether GFlowNets learn to sample from the target distribution remains underexplored. To tackle this issue, we first assess the extent to which a violation of the detailed balance of the underlying flow network might hamper the correctness of GFlowNet's sampling distribution. In particular, we demonstrate that the impact of an imbalanced edge on the model's accuracy is influenced by the total amount of flow passing through it and, as a consequence, is unevenly distributed across the network. We also argue that, depending on the parameterization, imbalance may be inevitable. In this regard, we consider the problem of sampling from distributions over graphs with GFlowNets parameterized by graph neural networks (GNNs) and show that the representation limits of GNNs delineate which distributions these GFlowNets can approximate. Lastly, we address these limitations by proposing a theoretically sound and computationally tractable metric for assessing GFlowNets, experimentally showing it is a better proxy for correctness than popular evaluation protocols.

## 1 INTRODUCTION

Generative flow networks (GFlowNets, Bengio et al., 2021; 2023) are reward-driven generative models for compositional objects (e.g., sequences or graphs) that have been successfully employed in several scientific domains (Deleu et al., 2022; 2023; da Silva et al., 2023; Zhang et al., 2023d; Jain et al., 2022; Bengio et al., 2021; Jain et al., 2023). In essence, GFlowNets cast sampling from an unnormalized distribution as solving a network flow problem (Bazaraa et al., 2004). Starting from an initial state, GFlowNets create valid samples by drawing a series of actions according to a (forward) policy network determined by the amount of flow between adjacent states. This process can be interpreted as spreading the total mass of the distribution (flow at the source) through trajectories that lead to elements in the target distribution's support (sink nodes).

While most works on GFlowNets are primarily empirical, developing a deeper theoretical understanding of GFlowNets is key to designing better models and assessment methodologies that are both theoretically sound and practically efficacious. In this regard, Bengio et al. (2023; 2021) laid out the technical foundations for GFlowNets, showing that a model satisfying the so-called *balance conditions* samples from the target discrete distribution. Lahlou (2023) extended this theory to the context of probability measures supported on arbitrary topological spaces. Also recently, the relationship of GFlowNets with variational inference (Malkin et al., 2023), reinforcement learning (Tiapkin et al., 2024), and diffusion models (Garipov et al., 2023; Lahlou, 2023) has been formally established.

Table 1: Main contributions of this work. Highlighted items represent methodological advancements.

| Section 3 | |
| --- | --- |
| Sensitivity to local failures for general SGs and targets | Thm. 1 |
| Formulation of weighted DB loss and comparison against DB | Eq. 6, Fig. 4 |
| **Section 4** | |
| Universal approximation of distributions over trees | Thm. 2 |
| Representational limits of 1-WL GFlowNets | Thm. 3 |
| Formulation of Look-Ahead (LA) GFlowNets | Eq. 7 |
| LA-GFlowNets $\succ$ Standard GFlowNets | Thm. 4, Fig. 6 |
| **Section 5** | |
| Definition of FCS as a tractable goodness-of-fit metric | Def. 1 |
| Relationship between FCS and TV | Thm. 5, Cor. 1 |
| FCS is highly correlated to TV | Sec. 5.1 |
| Inadequacy of commonly used evaluation protocols | Sec. 5.2 |

Despite these advances, a question of important practical implications remains: *when do a GFlowNet correctly learn its target distribution?* More specifically, little is known regarding the sensitivity of a GFlowNet's accuracy to balance violations, the possible causes of imbalance, or how to evaluate the distributional correctness of GFlowNets for large state spaces in a principled manner.

This paper establishes a series of results to address these fundamental questions. Firstly, we provide bounds on the total variation of GFlowNets as functions of balance fluctuations/violations. By considering tree-structured state graphs with identical rewards, we show that flow imbalances at different depths have a non-uniform impact on the approximation capabilities of GFlowNets — more specifically, balance mismatches near the root state may have a higher impact than those near terminal states. We also extend our analysis to show that similar results hold for general directed acyclic state graphs (DAGs) and multimodal target distributions. To illustrate the pragmatic benefits of these insights, we devise a novel family of learning objectives extending the traditional detailed balance loss. As we demonstrate in Section 3, this approach often accelerates training convergence.

After delving into the *consequences* of an imbalanced flow network on the GFlowNet's accuracy, we take a closer look at its potential *causes*. Towards this objective, we study the distributional limits of GFlowNets when sampling graph-structured objects. Notably, most applications of GFlowNets consist of sampling from distributions over graphs, which render graph neural networks (GNNs) (Gori et al., 2005; Gilmer et al., 2017; Xu et al., 2019) particularly convenient to parameterize policy networks. In fact, GNNs are often used to parameterize the policies in practice (Bengio et al., 2021; Roy et al., 2023; Zhu et al., 2023; Zhang et al., 2023d; Pandey et al., 2024). With this in mind, we provide constructions exposing their shortcomings. While GNN-based GFlowNets can express any distribution over trees under mild conditions, we show that there are simple state graphs and target distributions from which no GFlowNet can correctly sample. We leverage our analysis to introduce *look-ahead GFlowNets* (LA-GFlowNets), a simple yet effective scheme to provably boost the expressiveness of GFlowNets. In essence, LA-GFlowNet incorporates children-state embeddings as inputs to the forward policy. This allows LA-GFlowNets to distinguish actions that lead to distinguishable states but cannot be told apart by the Weisfeiler-Leman (WL) (Weisfeiler & Lehman, 1968) test.

Remarkably, these impossibility results underline the importance of having a reliable metric for probing the accuracy of a trained GFlowNet. In this sense, we also provide a theoretically sound framework for the distributional assessment (i.e., goodness-of-fit) of GFlowNets in high-dimensional state spaces, which we call *flow consistency in sub-graphs* (FCS) metric. Put simply, FCS consists of a Monte Carlo estimate of the average L1 error w.r.t. a distribution of "cuts" of the target's support. The FCS metric is a proxy for the absolute error between a GFlowNet's sampling and target distributions, and we empirically show that FCS highly correlates with the (often intractable) L1 error while requiring up to three orders of magnitude less compute. In contrast, we show that popular evaluation metrics do not accurately capture distributional correctness, e.g., the number of high-reward states visited during training and the average reward of the top-$k$ scoring states

(Jang et al., 2024; Kim et al., 2024; Bengio et al., 2021). In our view, this contribution is extremely valuable for ensuring smooth progress in the GFlowNet literature.

In Table 1, we summarize the main contributions of this work. Section 3 analyzes the distributional correctness of GFlowNets as a function of balance violations and leverages these theoretical insights to propose a family of *weighted detailed balance* (WDB) losses. Section 4 discusses the representational limits of GFlowNets for graph domains and proposes LA-GFlowNets to boost the expressive power of GNN-based GFlowNets. Finally, Section 5 proposes FCS as a theoretically grounded metric to assess the accuracy of GFlowNets well-suited to high-dimensional settings. Importantly, all sections of this work provide experiments to substantiate our theoretical analyses, illustrating the claims and demonstrating the practical relevance of the methodological contributions.

## 2 BACKGROUND

**Notations.** Let $\mathcal{X}$ be a finite set, which we call the set of *terminal states*, and $R$ be an unnormalized distribution over $\mathcal{X}$, which is also called a *reward function* (Bengio et al., 2021). We define the set of states $\mathcal{S}$ as an extension of $\mathcal{X}$ comprising two distinctive elements: an *initial state*, $s_o \in \mathcal{S}$, and a *final state*, $s_f \in \mathcal{S}$. We hence define a DAG $\mathcal{G} = (\mathcal{S}, \mathcal{E})$, termed *state graph* (SG), such that (i) there are no incoming edges to $s_o$; (ii) for each $x \in \mathcal{X}$, there is a directed path from $s_o$ to $x$; (iii) there is an edge from each $x$ to $s_f$, which is not directly connected to any other state in $\mathcal{S}$; and (iv) there are no outgoing edges from $s_f$. Figure 1 illustrates a state graph in which the states represent multisets and the edges denote the addition of an element. We say that a trajectory $\tau = (s_j)_{j=0}^h$ on $\mathcal{G}$ is *complete* if it starts at $s_o$ and ends at $s_f$,

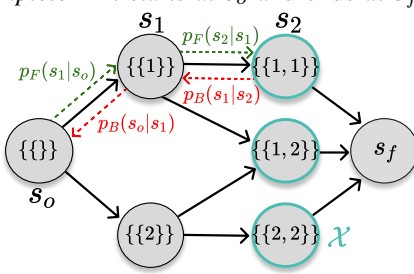

$s_1$

and write $\tau \rightsquigarrow x$ to denote a complete trajectory finished by the transition $(x, s_f)$; for example, $(s_o, s_1, s_2, s_f)$ in Figure 1 is a complete trajectory. A *forward policy* over $\mathcal{G}$ is a function $p_F \colon \mathcal{S} \times \mathcal{S} \to \mathbb{R}_+$ for which $p_F(s, \cdot)$ is a probability measure supported on $s$'s children in $\mathcal{G}$, denoted by $\mathrm{child}(s)$. We use $p_F(\cdot|s)$ and $p_F(s, \cdot)$ interchangeably. A *backward policy* $p_B$ is a forward policy over $\mathcal{G}$'s transpose. For a complete trajectory $\tau$, we write $p_F(\tau) = \prod_{i=1\ldots h} p_F(s_{i-1}, s_i)$. A *flow* is a function $F \colon \mathcal{S} \to \mathbb{R}_+$ s.t. $F|_{\mathcal{X}} = R$. We denote the cardinality operator as #. For a trajectory $\tau$, $\#\tau$ denotes its number of transitions.

Figure 1: Illustration of a state graph.

**GFlowNets.** A *GFlowNet* learns a forward policy $p_F$ and (sometimes) a backward policy $p_B$ and a flow function $F$ on a SG $\mathcal{G}$ such that the marginal distribution of $p_F$ over $\mathcal{X}$, $p_T(x) = \sum_{\tau \rightsquigarrow x} p_F(\tau)$, matches a given reward function $R$ (up to a normalizing factor). We refer to $p_T$ as the GFlowNet's *sampling distribution* and to $\pi \propto R$ as the reward-induced probability measure over $\mathcal{X}$. Also, we will often denote a GFlowNet by $(\mathcal{G}, p_F, p_B, F)$ or (when there is no risk of ambiguity) just $(p_F, p_B, F)$. In many applications, $p_F$ and $F$ are parameterized as a GNN (Bengio et al., 2021; Zhang et al., 2023d) or as a transformer (Deleu et al., 2022; Kim et al., 2024) and $p_B$ is fixed as a uniform policy. Since the seminal work of Bengio et al. (2021), numerous learning objectives have been proposed to estimate the model's parameters (Malkin et al., 2022; 2023; Madan et al., 2022; Zhang et al., 2023b), most of which are based on the principle of *network balance* — ensuring that the incoming and outgoing flows in a state are equal. The *detailed balance* (DB) loss, for instance, enforces the *detailed balance condition* $F(s)p_F(s'|s) = F(s')p_B(s|s')$ by minimizing the average log-squared difference between the incoming and outgoing flows of states within a trajectory (Bengio et al., 2023; Zhang et al., 2023d),

$$\mathcal{L}_{\mathrm{DB}}(p_F, p_B, F) = \mathbb{E}_\tau \left[ \frac{1}{\#\tau} \sum_{(s,s') \in \tau} \left( \log \frac{F(s)p_F(s'|s)}{F(s')p_B(s|s')} \right)^2 \right] \tag{1}$$

with the hard-coded constraint $F(s') = R(s')$ when $s' \in \mathcal{X}$ is a terminal state; the expectation is computed with respect to any positive probability measure over trajectories. Other popular learning objectives, such as the *trajectory balance* (TB) (Malkin et al., 2022) and *subtrajectory balance* (SubTB) (Madan et al., 2022) losses, are reviewed in the supplement.

**Assessment of GFlowNets.** For most problems, $\mathcal{X}$ is intractably large and it is not possible to directly compare the GFlowNet's sampling distribution to the target $R$. As a consequence, assessing the goodness-of-fit and convergence rate of GFlowNets is a challenging problem. To avoid this issue, a

common practice in the literature (Pan et al., 2023a;b; 2024; Zhang et al., 2023d; Jang et al., 2024) is to measure the count and average reward of the highest-scoring states found during training. The intuition is that a well-fitted model will quickly locate high-probability regions of the target distribution. More precisely, let $\mathcal{X}_T \subseteq \mathcal{X}$ be the terminal states found during training, $R_o \in \mathbb{R}_+$ be a hand-crafted threshold, and $\mathcal{H}(\mathcal{X}_T, R_o) = \{x \colon x \in \mathcal{X}_T \wedge R(x) \geq R_o\}$ be the samples in $\mathcal{X}_T$ with reward larger than $R_o$. Also, let $\{\!\{x_1, \ldots, x_S\}\!\} \sim p_T$ be $S$ samples from the trained GFlowNet. Then, we define

$$\text{Avg}(\mathcal{X}_T, R_o) = \sum_{x \in \mathcal{H}(\mathcal{X}_T, R_o)} \frac{R(x)}{\#\mathcal{H}(\mathcal{X}_T, R_o)} \text{ and } \text{Acc}(p_T) = \min\left\{ \frac{\frac{1}{S}\sum_{1 \leq i \leq S} R(x_i)}{\mathbb{E}_{x \sim R}[R(x)]}, 1 \right\}. \quad (2)$$

This second metric, referred to as *accuracy* by Shen et al. (2023); Kim et al. (2024), measures the proximity of the GFlowNet to the target based on the expected reward — assuming that $\mathbb{E}_{x \sim R}[R(x)]$ can be computed. Importantly, we will show in Section 5 that $\text{Avg}(\mathcal{X}_T, R_o)$, $\#\mathcal{H}(\mathcal{X}_T, R_o)$, and $\text{Acc}(p_T)$ are not necessarily connected to the closeness of a GFlowNet to the global minimum of its learning objective and, therefore, may lead to misguided conclusions if not interpreted carefully.

**The expressive power of GNNs and the 1-WL isomorphism test.** Graph neural networks (GNNs) are the dominating paradigm for graph representation learning (Xu et al., 2019; Hamilton, 2020; Wang et al., 2024; Corso et al., 2024). Most GNNs employ a multi-layered message-passing scheme, interleaving neighborhood aggregation and update operations at each layer. Specifically, for each node $v$ at layer $\ell$, the aggregation is a nonlinear function of the $(\ell - 1)$-layer representations of $v$'s neighbors. The update step computes a new representation for $v$ based on its representation at layer $\ell - 1$ and the aggregated messages (output of the aggregation step). Importantly, message-passing GNNs have well-known representational limits, which are upper-bounded by the first-order Weisfeiler-Lehman isomorphism test (1-WL) (Xu et al., 2019; Weisfeiler & Lehman, 1968). We provide more details regarding this relationship in Appendix B.

## 3 ON THE PROPAGATION OF ERRORS IN FLOW NETWORKS

Our investigation starts with the main source of distributional errors in a GFlowNet, namely, the lack of balance in the underlying flow network. In this pursuit, the primary question we wish to address is: what is the impact of violations to the balance conditions on the goodness-of-fit of GFlowNets?

### 3.1 BOUNDS ON THE TOTAL VARIATION OF GFLOWNETS

To build intuition, our first result (Remark 1) quantifies the extent to which a violation to the detailed balance condition in a single node might affect the TV distance between the GFlowNet's sampling distribution and a uniform target in the case of tree-structured SGs, which are often featured in applications, e.g., (Jain et al., 2022; Jiralerspong et al., 2023; Liu & et al., 2023; Hu et al., 2023).

*Remark* 1 (TV for tree-structured SGs). Let $(\mathcal{G}, p_F, p_B, F)$ be a GFlowNet balanced with respect to a reward $R$, where $\mathcal{G}$ is a directed regular tree with branching factor $g$ and depth $h$, and $R$ is unnormalized uniform. Also, consider the GFlowNet $(\mathcal{G}, \tilde{p}_F, p_B, \tilde{F})$ such that i) $\tilde{F}(s_o) = F(s_o) + \delta$ and $\tilde{F}(s^\star) = F(s^\star) + \delta$ for some $s^\star \in \text{child}(s_o)$ and $\delta \geq 0$; ii) $\tilde{F}(s) = F(s)$ for all $s$ not reachable from $s^\star$; iii) $\tilde{F}(s) = \sum_{s' \in \text{child}(s)} \tilde{F}(s')$; and iv) $\tilde{p}_F(s, s') \propto \tilde{F}(s') \; \forall (s, s') \in \mathcal{E}(\mathcal{G})$ (see Figure 2). Let $\tilde{p}_T$ be the marginal distribution induced by $\tilde{p}_F$. The TV between $\tilde{p}_T$ and $\pi \propto R$ satisfies

$$\epsilon\left(\delta, g, F(s_0)\right) \leq \text{TV}\left(\tilde{p}_T, \pi\right) \leq \epsilon\left(\delta, g^h, F(s_0)\right), \text{ with } \epsilon(\delta, x, t) := (1 - 1/x)\,\delta/t + \delta. \quad (3)$$

Naturally, the upper and lower bounds are increasing functions of $\delta$. Importantly, these bounds are tight, i.e., for any $\delta$, there is a corresponding flow function for which the TV equals the stated bounds. Also, we note that the upper bound $\epsilon(\delta, g^h, F(s_o))$ increases monotonically with the number of leaves $g^h$, i.e., the further the imbalanced edge $(s_o, s^\star)$ is from the leaves, the greater the potential damage to accuracy. Notably, this demonstrates that the effect of balance violations on the distributional approximation is heterogeneously spread among the SG's edges. We experimentally validate these findings for the benchmark tasks outlined in Section 2 in Figure 3.

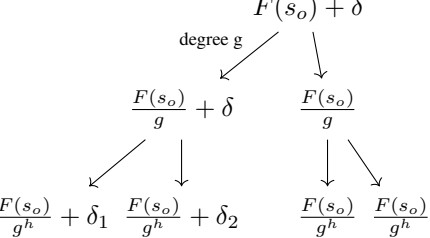

Figure 2: Tree-structured SG w/ excess flow $\delta$ from $s_0$ to left child. We omit node labels.

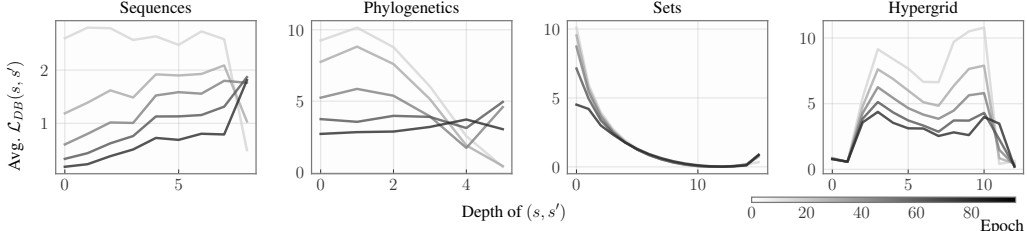

Figure 3: **Average** $\mathcal{L}_{\text{DB}}(s, s') := (\log(F(s)p_F(s, s') - \log(F(s')p_B(s, s'))))^2$ **along randomly sampled trajectories** during the early stages of training. As suggested by our analysis, the DB loss is unevenly distributed across a trajectory, with different transitions influencing the loss in diverse ways.

As we shall show in Theorem 1, these intuitive results can be extended to the context of arbitrarily shaped SGs labeled with any target probability measure $\pi$. In this broader setting, of which Equation 3 is a particular case, the tree-inherited concepts of *depth* and *branching factor* of a state $s$ are replaced by the *total probability mass* accumulated by the terminal descendants of $s$. The reader is invited to observe that, under the assumptions of Remark 1, these properties are interchangeable. From a practitioner's perspective, however, the exact computation of the quantities appearing in Theorem 1 is unfeasible for most benchmark and realistic problems. Consequently, our empirical analysis in the following section leverages the insights from Theorem 1 and the computational tractability of Remark 1 to derive a *weighted detailed balance* (WDB) loss which, by assigning different weights to different transitions based on their distance to the initial state of the SG, aims at facilitating the search for a balanced flow assignment and speeding up the training convergence.

**Theorem 1** (TV bounds for arbitrary distributions). *Let* $(\mathcal{G}, p_F, p_B, F)$ *be a GFlowNet with arbitrary state graph* $\mathcal{G}$ *satisfying the DB condition w.r.t. an arbitrary reward* $R$. *Similarly to Remark 1, define* $(\mathcal{G}, \tilde{p}_F, p_B, \tilde{F})$ *by increasing the flow* $F(s)$ *in some node* $s$ *by* $\delta$ *and redirecting the extra flow to a direct child* $s^\star$ *by properly adjusting* $p_F(s, \cdot)$. *Likewise,* $\tilde{F}$ *is defined by propagating the extra flows to all states reachable from* $s^\star$. *Also, let* $\mathcal{D}_{s^\star} \subseteq \mathcal{X}$ *be the set of terminal states reachable from* $s^\star$. *Then, the TV between the distribution* $\tilde{p}_T$ *over* $\mathcal{X}$ *induced by* $\tilde{p}_F$ *and the normalized target* $\pi \propto R$ *satisfies*

$$\frac{\delta}{F(s_0) + \delta} \left( 1 - \sum_{x \in \mathcal{D}_{s^\star}} \pi(x) \right) \leq \text{TV}(\tilde{p}_T, \pi) \leq \frac{\delta}{F(s_0) + \delta} \left( 1 - \min_{x \in \mathcal{D}_{s^\star}} \pi(x) \right). \qquad (4)$$

## 3.2 APPLICATION TO GFLOWNET TRAINING

**Weighted DB.** We note that, by default, the DB loss in Equation 1 computes an arithmetic average of the transition-level errors. Intrinsically, this design encodes that each transition has the same impact on our overall goal of approximating the target distribution. Nonetheless, as indicated by our theoretical and empirical analyses in Section 3.1, this is not the case. Therefore, we construct a family of *weighted detailed balance* (WDB) losses,

$$\mathcal{F}_{\text{WDB}} = \left\{ \mathcal{L}_\gamma(s, s') : (s, s') \mapsto \gamma(s, s') \left( \log \frac{F(s)p_F(s'|s)}{F(s')p_B(s|s')} \right)^2 \middle| \gamma : \mathcal{S} \times \mathcal{S} \to \mathbb{R}_+ \right\}, \qquad (5)$$

and train a GFlowNet by choosing a $\mathcal{L}_\gamma \in \mathcal{F}_{\text{WDB}}$ and minimizing the stochastic objective

$$\mathcal{L}_{\text{WDB}}^\gamma(p_F, p_B, F) := \mathbb{E}_\tau \left[ \frac{1}{\sum_{(s,s') \in \tau} \gamma(s, s')} \sum_{(s,s') \in \tau} \mathcal{L}_\gamma(s, s') \right]. \qquad (6)$$

We are left with the task of choosing an appropriate $\gamma$. Inspired by recent advances in diffusion probabilistic models (Kingma et al., 2021; Kingma & Gao, 2023), we might choose a $\gamma$ ensuring that no term in Equation 6 dominates the loss. In light of Remark 1, any monotonically decreasing function on $\#\mathcal{D}_{s'}$ (i.e., the number of terminal descendants of $s'$) would be a principled choice for $\gamma$. Here, we use $\gamma(s, s') = 1/\#\mathcal{D}_{s'}$, but acknowledge that other $\gamma$ might be optimal for different tasks.

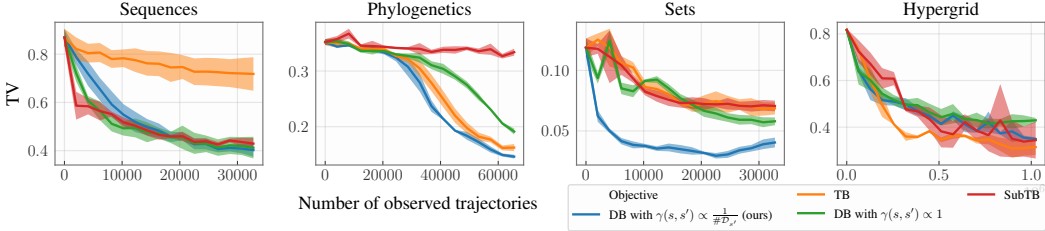

Figure 4: **WDB** performs competitively with or better than **DB**, **SubTB**, and **TB**. By weighting each $(s, s')$ in inverse proportion to the number of terminal descendants of $s'$ (i.e., $\gamma(s, s') = 1/\#\mathcal{D}_{s'}$) in the DB loss, a faster convergence in terms of TV w.r.t. the standard objective is achieved.

**Empirical illustration.** We compare the performance of WDB against TB, SubTB, and the standard DB objective (with $\gamma \equiv 1$) in Figure 4 using four benchmark tasks for GFlowNets: autoregressive sequence design (Jain et al., 2022; Malkin et al., 2022; Jiralerspong et al., 2023), phylogenetic inference (Zhou et al., 2024), set generation (Bengio et al., 2023; Pan et al., 2023b;a; Jang et al., 2024), and hypergrid navigation (Malkin et al., 2022; 2023; Madan et al., 2022). We provide more details regarding the experimental setup in Appendix E. Remarkably, using WDB leads to faster convergence of the GFlowNet's sampling distribution to the target distribution for phylogenetic inference and set generation — measured in terms of the TV distance. Note these two environments are exactly the ones for which early-stage transitions dominate the loss, as shown in Figure 3. For the two remaining cases, WDB performs approximately on par with standard DB. These results, which are discussed at length in Section E.4 (see Figure 11), suggest that one may drastically improve the DB loss by adequately weighting each transition within a trajectory.

## 4 DISTRIBUTIONAL LIMITS OF GNN-BASED POLICY NETWORKS

Our analysis so far has focused on the impact of imbalanced nodes on the GFlowNet's sampling distribution. We now step back and analyze a natural cause for this lack of balance: parametrization. Notably, some of the hottest applications of GFlowNets lie in graph domains and leverage GNNs to incorporate desirable inductive biases (e.g., Bengio et al., 2021; Nica et al., 2022; Roy et al., 2023; Zhang et al., 2023d; Zhu et al., 2023; Pandey et al., 2024). Thus, this section explores the representational limits of GNN-based GFlowNets. Towards this goal, we show their universal capacity of approximating distributions over trees. Then, we construct a family of problems that a GFlowNet based on 1-WL GNNs, termed as *1-WL GFlowNet*, cannot solve, showing that balance violations may arise due to limited expressivity of the policy network.

Our first result (Theorem 2) demonstrates that, for any reward supported over trees, there is an 1-WL GFlowNet capable of sampling proportional to that reward. To achieve this result, we construct a simple generative process starting from a totally disconnected graph and adding one edge at a time, always yielding only one non-singleton component.

**Theorem 2** (Universality of 1-WL GFlowNets for trees.). *If $\mathcal{S}$ is a collection of trees such that $(s, s') \in \mathcal{E}$ implies that $s \subset s'$ ($s$ is a proper subtree of $s'$) with $\#E(s') = \#E(s) + 1$, then there is a GFlowNet equipped with 1-WL GNNs can approximate any distribution $\pi$ over $\mathcal{X} \subseteq \mathcal{S}$.*

Theorem 2 certifies that 1-WL GFlowNets can sample from arbitrary distributions over trees. However, the 1-WL test is not a perfect oracle for isomorphism. A natural question is: *are there limits to the representational power of 1-WL GFlowNets?* Theorem 3 shows a broad family of cases (i.e., combinations of SGs and reward functions) for which 1-WL GFlowNets fail. This result rests on the fact that states must distribute flow evenly to children if the actions leading to them are 1-WL indistinguishable.

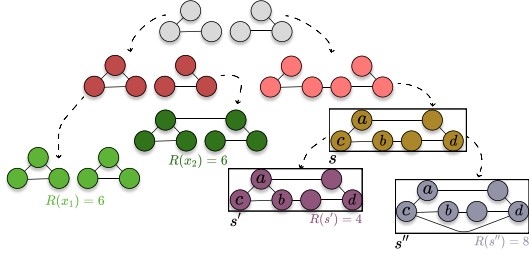

Figure 5: A combination of a state graph and reward function that causes 1-WL GFlowNets to fail.

**Illustrating failure modes of 1-WL GFlowNets.** Figure 5 provides a construction where 1-WL GFlowNets fails to achieve balance. Note that the actions leading to the children $s'$ and $s''$ of $s$ (enclosed by a box) are 1-WL indistinguishable. Hence, 1-WL policies distribute the flow in $s$ equally among $s'$ and $s''$, failing to match the distinct rewards $R(s') = 4$ and $R(s'') = 8$. Theorem 3 extends this intuition to a wider range of cases.

**Theorem 3** (Limitations of GNN-based GFlowNets). *Let $\mathcal{G} = (\mathcal{S}, A)$ be a state graph and $R : \mathcal{X} \subseteq \mathcal{S} \to \mathbb{R}^+$ be a reward function. Suppose $\mathcal{G}$ is a directed tree. Let $T(s) \subseteq \mathcal{X}$ for $s \in \mathcal{S}$ denote the set of terminal states reachable by a directed path starting at $s$. If there is a state $s = (V, E) \in \mathcal{S}$ and two pairs of nodes $(a, b) \neq (c, d) \in V^2 \setminus E$ that are not 1-WL distinguishable and $\sum_{x \in T(s')} R(x) \neq \sum_{x \in T(s'')} R(x)$ with $s' = (V, E \cup \{(a, b)\})$ and $s'' = (V, E \cup \{(c, d)\})$ (illustrated in Figure 5), then there is no 1-WL GFlowNet capable of approximating $\pi \propto R$ with TV zero.*

We now leverage these insights to propose a more expressive GNN-based GFlowNet: *Look-ahead GFlowNets* (LA-GFlowNets). The rationale of LA-GFlowNets is to incorporate children's graph embeddings as inputs to the forward policy. This allows LA-GFlowNets to disambiguate between children states obtained from 1-WL equivalent actions, enabling assignment of uneven probabilities to non-distinguishable actions as long as the embeddings of corresponding children states differ.

More formally, let $s'$ and $s$ be two neighboring nodes in the SG, differing only by an edge $(u, v)$ not in $s$ — recall that $s$ and $s'$ are graphs themselves. Let also $\phi_{v|G}$ be the 1-WL embedding of a node $v$ within a graph $G$. Then, LA-GFlowNets' forward policy can be described as

$$p_F(s, s') \propto \exp\left\{\text{MLP}\left(\psi_1\left(\{\phi_{u|s}, \phi_{v|s}\}\right) \parallel \psi_2\left(\{\phi_{w|s'}\}_{w \in V(s')}\right)\right)\right\}, \qquad (7)$$

where $\psi_1$ and $\psi_2$ are order-invariant functions (Zaheer et al., 2017). Since child embeddings are added (via concatenation) to the original action embedding, there is no loss of expressiveness w.r.t. 1-WL GFlowNets. On the other hand, LA-GFlowNets can perfectly approximate cases like the one depicted in Figure 5. Theorem 4 states the superior expressiveness of LA-GFlowNets.

**Theorem 4** (LA-GFlowNet is more expressive than 1-WL GFlowNet). *If there is a 1-WL forward policy inducing a sampling distribution proportional to a reward $R$, there is a LA-GFlowNet forward policy over the same SG with a sampling distribution proportional to R. The converse does not hold.*

**Empirical illustration.** To demonstrate the limitations of 1-WL GFlowNets, we define next a group $\mathcal{G}$ of SGs for which there are actions that, despite leading to non-isomorphic states, cannot be distinguished by a GNN-based policy. In this scenario, let $\mathcal{R}_{n,k}$ be the set of regular graphs with $n$ nodes of degree $k$. Then, let $\mathcal{G}$ be the set of SGs $C_1 \leftarrow P \to C_2$ such that $P \in \mathcal{R}_{n,k}$ and $C_1$ and $C_2$ are non-isomorphic graphs differing from $P$ by a single additional edge; see Figure 9 in the supplement for an illustration. Note that, due to the (graph-theoretic) regularity of $P$, $p_F(P, C_1) = p_F(P, C_2)$ for any GNN-based $p_F$. Thus, the corresponding GFlowNet is inherently unable to learn a non-uniform distribution on $\{C_1, C_2\}$. LA-GFlowNets, in contrast, are not constrained by such limited expressivity. As an example, we create four triples $(C_1, P, C_2)$ with $n = 8$, $k = 3$, $R(C_1) = 0.1$ and $R(C_2) = 0.9$. Under these conditions, Figure 6 shows LA-GFlowNet can accurately sample from the target distribution. However, a standard GNN-based GFlowNet can only sample from a uniform, attaining a (constant) $L_1$ error of $0.4$ throughout training.

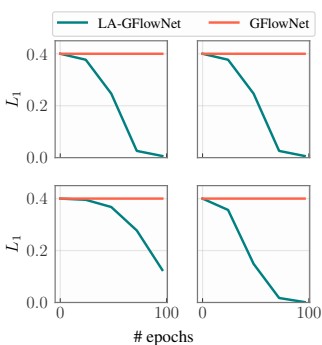

Figure 6: Illustrations in which LA-GFlowNets succeed but standard GNN-based GFlowNet fail.

## 5 CONVERGENCE DIAGNOSTICS FOR GFLOWNETS

Finally, with the understanding that there are distributions from which a GFlowNet cannot sample, we ask: how can we tractably assess the closeness of a GFlowNet's to its target? To answer this, we propose a provably correct and computationally amenable metric for probing the distributional incorrectness of GFlowNets (Section 5.1), termed *Flow Consistency in Subgraphs* (FCS). Strikingly, we compare FCS against three popular techniques for assessing the convergence of GFlowNets, namely, the number of modes, average reward of top-scoring samples, and Shen's accuracy, and show that FCS is often the only metric accurately reflecting a GFlowNet's goodness-of-fit (Section 5.2).

## 5.1 PROBING GFLOWNETS' DISTRIBUTIONAL INCORRECTNESS

**Flow Consistency in Subgraphs (FCS).** The basic principle of FCS is to estimate the discrepancy between ratios of probabilities (Hyvärinen, 2007) instead of measuring the divergence between the intractable learned and target distributions. For this, we recall that the marginal distribution $p_T$ of a GFlowNet $(\mathcal{G}, p_F, p_B)$ over the terminal states $\mathcal{X}$ can be computed as

$$p_T(x) := \sum_{\tau:\, s_o \rightsquigarrow x} p_F(\tau) = \mathbb{E}_{\tau \sim p_B(\cdot|x)}\left[\frac{p_F(\tau)}{p_B(\tau|x)}\right] \tag{8}$$

for each $x \in \mathcal{X}$. For most benchmark tasks, e.g., hypergrid environment (Malkin et al., 2023), set generation (Shen et al., 2023), and sequence design (Jain et al., 2022), we can exactly and tractably compute $p_T$. In autoregressive problems (Jain et al., 2022), for instance, there is a single trajectory $\tau_x$ leading to each $x \in \mathcal{X}$. Hence, $p_T(x) = p_F(\tau_x)$ can be directly evaluated. Similarly, for small state graphs, the sum in Equation 8 can be explicitly calculated by enumerating the trajectories $\tau$ finishing at $x$. When exact computation of $p_T$ is unfeasible, a Monte Carlo estimator of the expectation in Equation 8 offers an accurate approximation. In this case, FCS consists of comparing restrictions of $p_T(x)$ and $R(x)$ to *random subsets* of $\mathcal{X}$. We formalize this procedure in the definition below.

**Definition 1** (Flow Consistency in Sub-graphs). Let $P_S$ be a positive probability distribution on $\beta$-sized subsets of $\mathcal{X}$, $\beta \geq 2$. For each $S \subseteq \mathcal{X}$, define the restrictions of $p_T$ and $R$ to the set $S$ as

$$p_T^{(S)}(x) = \frac{\mathbf{1}_{\{x \in S\}} p_T(x)}{\sum_{y \in S} p_T(y)} \text{ and } R^{(S)}(x) = \frac{\mathbf{1}_{\{x \in S\}} R(x)}{\sum_{y \in S} R(y)} \text{ for } x \in \mathcal{X}. \tag{9}$$

We define FCS as the expected TV between $p_T^{(S)}$ and $R^{(S)}$:

$$\text{FCS}(p_T, R) := \mathbb{E}_{S \sim P_S}[\text{TV}(p_T^{(S)}, R^{(S)})]. \tag{10}$$

Clearly, $\text{FCS}(p_T, R) \in [0, 1]$. Moreover, Theorem 5 shows that $\text{FCS}(p_T, R) = 0$ only if $p_T(x) \propto R(x)$, asserting the conceptual correctness of our metric.

**Theorem 5** (Equivalence between TV & FCS). *Let $P_S$ be any full-support distribution over $\{S \subseteq \mathcal{X} : \#S = \beta\}$ for some $\beta \geq 2$. Also, let $\text{TV}(p_T, \pi) = {}^1\!/_2 \sum_{x \in \mathcal{X}} |p_T(x) - \pi(x)|$ be the TV distance between $p_T$ and $\pi := R/Z$, $Z = \sum_{x \in \mathcal{X}} R(x)$. Then, $\text{TV}(p_T, \pi) = 0$ if and only if $\text{FCS}(p_T, R) = 0$.*

Notably, $\beta$ interpolates FCS between a ratio-matching-like metric (Hyvärinen, 2007) ($\beta = 2$) and the TV distance ($\beta = \#\mathcal{X}$). Here, we set $\beta$ as the size of the batch of trajectories used in training. In this respect, Corollary 1 clarifies the role of $\beta$ in FCS in terms of its proximity to the TV distance.

*Corollary* 1 (Role of $\beta$ in FCS). Let $P_S(S; \beta) = \mathbf{1}_{\{\#S = \beta\}} \binom{n-1}{\beta-1}^{-1} \sum_{x \in S} p_T(x)$ be a distribution over $\beta$-sized subsets of $\mathcal{X}$. Also, let $p_T(S) = \sum_{x \in S} p_T(x)$, and define $\pi(S)$ similarly. Then,

$$\left| \text{TV}(p_T, \pi) - \mathbb{E}_{S \sim P_S(\cdot; \beta)}\left[\text{TV}\left(p_T^{(S)}, R^{(S)}\right)\right] \right| \leq \frac{1}{2} \cdot \frac{\#\mathcal{X}}{\beta} \cdot \max_{S \subseteq \mathcal{X}, \#S = \beta} |p_T(S) - \pi(S)|. \tag{11}$$

**An implementation of FCS.** First, we emphasize that FCS can be easily extended to accommodate variably sized subsets of $\mathcal{X}$. To see this, let $P_S$ be any positive distribution in $\{S \subseteq \mathcal{X} : \#S \leq \beta\}$ and note $\mathbb{E}_{S \sim P_S}[TV(p_T^{(S)}, R^{(S)})] = 0$ only if $\text{FCS}(p_T, R) := \mathbb{E}_{S \sim P_S(\cdot | \#S = \beta)}[TV(p_T^{(S)}, R^{(S)})] = 0$. In this context, our implementation of FCS defines $P_S$ as the distribution over at-most-$\beta$-sized subsets of $\mathcal{X}$ corresponding to the terminal states of a batch of trajectories sampled from a fixed policy.

**PAC statistical guarantees for FCS.** From a statistical viewpoint, FCS approximates the distributional accuracy of a GFlowNet by probing the model on a relatively small fraction of the state graph. One might wonder, under these conditions, how the empirical estimate compares to a deterministic goodness-of-fit measure. Corollary 2 addresses this issue from a *probably approximately correct* (PAC) perspective, showing that an estimate of FCS closely approximates TV when a sufficiently large number of subsets is sampled (large $m$) and the model is relatively accurate (small error).

*Corollary* 2 (PAC bound for FCS). Let $P_S$ be as in Theorem 5. Then, for any $\delta \in (0, 1)$, with probability at least $1 - \delta$ over choosing $m$ i.i.d. $\beta$-sized subsets $S_1, \ldots, S_m \sim P_S$ of $\mathcal{X}$:

$$\text{TV}(p_T, \pi) \leq \frac{1}{m} \sum_{1 \leq i \leq m} \text{TV}\left(p_T^{(S_m)}, R^{(S_m)}\right) + \frac{\#\mathcal{X}}{2\beta} \cdot \max_{S \subseteq \mathcal{X}, \#S = \beta} |p_T(S) - \pi(S)| + \sqrt{2 \log \tfrac{1}{\delta}/m}.$$

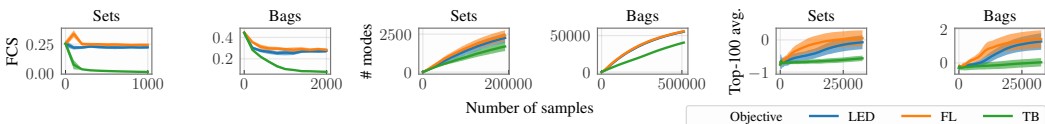

Figure 7: **FCS is a computationally feasible surrogate for the TV distance**. (left) FCS accurately represents TV in the considered tasks (right) while being up to three orders of magnitude faster to compute.

Figure 8: **FCS is the only metric correctly reflecting GFlowNet's distributional accuracy**. Number of modes (columns 3-4) and average reward (columns 5-6) of the highest scoring samples found during training do not accurately reflect GFlowNet's goodness-of-fit, while FCS does (columns 1-2). Remarkably, we consider the *terminally unrestricted* variants of LED- and FL-GFlowNets (see Proposition 1).

**Empirical illustration.** Figure 7 (left) shows that FCS closely resembles the TV distance for the tasks of set and sequence generation. In fact, the Spearman correlation between FCS and TV for these tasks is 0.99 and 0.90, respectively. Importantly, however, the estimation of FCS is up to three orders of magnitude faster than the computation of the TV distance (Figure 7 (right)). Remarkably, these results attest the usefulness of FCS as a general-purpose goodness-of-fit metric for GFlowNets.

## 5.2 CASE STUDY: LED- AND FL-GFLOWNETS WITH UNRESTRICTED FLOWS

**LED- and FL-GFlowNets.** When training GFlowNets, the learning signal is *sparse*; it is only available at the end of each trajectory via the reward function. LED- (Jang et al., 2024) and FL- (Pan et al., 2023a) GFlowNets aim at reducing this issue by reparametrizing $\log F(s, s')$ as the residual of a potential function $\phi_\theta(s, s')$, i.e., $\log F(s, s') = \phi_\theta(s, s') + \log \tilde{F}(s, s')$, and minimizing

$$\mathcal{L}_{\text{LED}}(s, s') = \left(\log \tilde{F}(s) + \log p_F(s, s') - \log p_B(s', s) - \log \tilde{F}(s') + \phi_\theta(s, s')\right)^2 \qquad (12)$$

for every $(s, s')$. For FL-GFlowNet, $\phi_\theta(s, s') = \xi(s') - \xi(s)$ is fixed as the gap between hand-crafted energy functions satisfying $\xi(x) = -\log R(x)$ for $x \in \mathcal{X}$ and $\xi(s_o) = 0$ (Pan et al., 2023a, Eq. (5)). For LED-GFlowNet, $\phi_\theta(s, s')$ is learned. Readers may consult Appendix C for further details.

**LED- and FL-GFlowNets with unrestricted flows.** Our findings below reveal that, even when the terminal flows $F(x)$ for $x \in \mathcal{X}$ are not constrained to equal $R(x)$, both LED- and FL-GFlowNets greatly outperform a standard GFlowNet according to conventional metrics. As we show both theoretically (Proposition 1) and empirically (Figure 8), however, constraining $F(x)$ to $R(x)$ is necessary to ensure GFlowNet's sampling correctness even under w.r.t.like parameterization. Importantly, we have significant reasons to believe that an unrestricted $F(x)$ was a part of some experiments in the original works of Pan et al. (2023a) and Jang et al. (2024) (see Section E.3), a fact that strengthens the need for a standard, easy-to-compute, and sound metric for GFlowNet assessment, such as FCS.

**Proposition 1** (Unpredictability of GFlowNets with unrestricted terminal flows). *Consider an FL- or LED-GFlowNet achieving $\mathcal{L}_{\text{LED}}(s, s') = 0$ for all transitions $(s, s')$ and trajectories $\tau$. Then, the learned marginal distribution over $\mathcal{X}$ satisfies $p_T(x) \propto R(x)\tilde{F}(x)$ for every $x \in \mathcal{X}$.*

We refer to GFlowNets trained without enforcing $F(x){=}R(x)$ as *terminally unrestricted* (TU).

**Experimental setup.** We empirically demonstrate that FCS is the *only metric* able to represent GFlowNet's accuracy when compared to three popular alternatives for assessing these models: number of modes, average reward of top-scoring samples, and Shen's accuracy (see Equation 2; Shen et al., 2023). For this, we consider the standard tasks of set and bag (multiset) generation, also featured in Jang et al. (2024)'s experimental campaign, and design of DNA sequences bindable with a

yeast PHO4 transcription factor. For the latter, we omit results for the FL-GFlowNet due to the lack of a clear candidate for $\xi$; results are presented in Figure 13. Appendix E contains further details.

**Results.** There are three main takeaways from Figure 8 and Table 2. First, our baseline model (TB-GFlowNet) accurately learns to sample from the target distribution (Figure 8, left), whereas the (terminally unrestricted) LED- and FL-GFlowNets variants do not. Second, both LED- and FL-GFlowNets find a significantly more valuable portion of the state space than their standard counterpart during training (Figure 8, middle, right), but fail to sample correctly. The large variance of the reported results is a consequence of the lack of a unique stationary solution to the models' respective learning objectives, as stated in Proposition 1, and was also observed by Pan et al. (2023a, Figure 2). Third, Shen's accuracy is not an appropriate proxy for goodness-of-fit. All in all, our experiments show that usual metrics should be used carefully when comparing the convergence speed of GFlowNets.

In contrast to conventional wisdom in the literature, these quantities do not directly measure a GFlowNet's closeness to a global minimizer of its learning objective. In view of this, our analyses highlight the importance of having a theoretically sound and computationally amenable metric for assessing GFlowNets to drive progress in the field. Strikingly, FCS is — as far as we know — the only alternative satisfying both of these constraints.

Table 2: **Shen et al. (2023)'s accuracy metric** incorrectly assigns perfect score to the provably unsound TU-FL and TU-LED GFlowNet's variants.

|      | LED | FL | TB |
|------|-----|-----|-----|
| Sets | $100.00_{\pm 0.00}$ | $100.00_{\pm 0.00}$ | $93.74_{\pm 0.98}$ |
| Bags | $100.00_{\pm 0.00}$ | $100.00_{\pm 0.00}$ | $81.38_{\pm 6.86}$ |
| PHO4 | $100.00_{\pm 0.00}$ | NA | $96.98_{\pm 0.77}$ |

## 6 CONCLUSIONS, LIMITATIONS, AND BROADER IMPACT

**Conclusions.** The learning objective of GFlowNets is to find a balanced flow assignment in a flow network. As such, the inaccuracies of a trained model are a consequence of violations to the posited balance conditions. In this work, we first argued that the impact of an imbalanced node on the GFlowNet's distributional correctness is heterogeneously distributed across the flow network. As a consequence, we extended the DB loss by non-uniformly weighting the transition-wise terms to account for this heterogeneity, which was shown to be effective in practice. For graph-structured generative tasks, we proved that these violations to the balance might be associated to the limited expressiveness of the GNN that parameterizes the policy network, which limits the range of distributions that a GFlowNet can sample from. To mitigate this issue, we introduced LA-GFlowNets to boost the expressive power of GNN-based GFlowNets by incorporating the embeddings of the children of a state into the policy network. Finally, we proposed FCS as a computationally amenable metric for probing the goodness-of-fit of GFlowNets to its target distribution when the learned flow assignment is potentially imbalanced. Notably, our experiments demonstrated that FCS is a better proxy to the GFlowNet's distributional accuracy than conventional diagnostic methods.

**Limitations.** Although our empirical analysis comprehends standard benchmark problems in the GFlowNet literature and is on par with other works in terms of the variety of generative tasks, it does not consider specialized applications such as molecule generation (Pandey et al., 2024) and natural language processing (Hu et al., 2023). Additionally, our weighting scheme for the WDB learning objective was heuristically derived. Albeit effective, it could likely be improved by taking into account desirable properties of a loss function, e.g., low gradient variance (Richter et al., 2020; Malkin et al., 2023). Lastly, we established results concerning the capabilities and limitations of 1-WL GNN-based GFlowNets due to their widespread use in practice (Kipf & Welling, 2017; Veličković et al., 2018; Xu et al., 2019; Corso et al., 2024). However, extending this analysis to more expressive (e.g., higher-order) GNNs is a promising direction (Morris et al., 2021).

**Broader impact.** In conclusion, we believe our work paves the road for many advancements in the GFlowNet literature. Firstly, the development of more effective weighting schemes for the DB objective in Section 3 may speed up GFlowNet training. Similarly, the limited expressivity of GNN-based GFlowNets laid out in Section 4 is a cautionary tale for using equivariant neural networks when parameterizing the model's policies and may be a useful tool for explaining the difficulty in approximating certain distributions, as well as developing more expressive models. Last but not least, we expect FCS to greatly impact the assessment of GFlowNets and the validation of novel learning objectives.

ACKNOWLEDGEMENTS

This work was supported by the Fundação Carlos Chagas Filho de Amparo à Pesquisa do Estado do Rio de Janeiro FAPERJ (SEI-260003/000709/2023), the São Paulo Research Foundation FAPESP (2023/00815-6), the Conselho Nacional de Desenvolvimento Científico e Tecnológico CNPq (404336/2023-0), and the Silicon Valley Community Foundation through the University Blockchain Research Initiative (Grant #2022-199610).

We acknowledge the Aalto Science-IT Project from Computer Science IT and FGV TIC for providing computational resources.

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

# A    RELATED WORKS

**GFlowNets.** Generative Flow Networks (GFlowNets) (Bengio et al., 2021; 2023; Lahlou, 2023) were proposed as an alternative to Monte Carlo Tree Search (Buesing et al., 2020) in DAG-structured environments. Presently, GFlowNets have been successfully applied with remarkable success in difficult problems such as causal discovery (Deleu et al., 2022; 2023; da Silva et al., 2023), molecule discovery (Bengio et al., 2021; Nica et al., 2022; Atanackovic & Bengio, 2024; Pandey et al., 2024), text infilling (Hu et al., 2023; Venkatraman et al., 2024), phylogenetic inference (Zhou et al., 2024), and combinatorial optimization (Zhang et al., 2023b;c). Similarly, a large effort has been devoted to the development of easier-to-minimize learning objectives with enhanced credit assignment (Bengio et al., 2023; Malkin et al., 2022; Madan et al., 2022; Pan et al., 2023b; Tiapkin et al., 2024) and better exploratory policies (Pan et al., 2023a; Vemgal et al., 2023; Rector-Brooks et al., 2023; Lau et al., 2023; 2024; Jang et al., 2024; Kim et al., 2024). In our work, we considered both the trajectory balance,

$$\mathcal{L}_{\text{TB}}(p_F, p_B, F) = \mathbb{E}_\tau \left[ \left( \log \frac{F(s_o)p_F(\tau)}{p_B(\tau|x)R(x)} \right)^2 \right], \tag{13}$$

and the subtrajectory balance ($\tau_i$ denotes the $i$th state of $\tau$ and $\#\tau+1$, the number of states in $\tau$),

$$\mathcal{L}_{\text{SubTB}}(p_F, p_B, F) = \mathbb{E}_\tau \left[ \sum_{1 \le i < j \le \#\tau+1} \frac{\lambda^{j-i}}{\sum_{1 \le s < t \le \#\tau+1} \lambda^{t-s}} \left( \log \frac{F(\tau_i)p_F(\tau_{i:j}|\tau_i)}{p_B(\tau_{i:j}|\tau_j)F(\tau_j)} \right)^2 \right], \tag{14}$$

which are the most popular loss functions for training GFlowNets. Nonetheless, the problems of identifying the failure modes and soundly assessing the accuracy of GFlowNets have received far less attention from the literature, with Shen et al. (2023)'s work being the most closely related to ours.

**Expressiveness of GNNs.** Graph neural networks (GNNs) are the leading approach for representation and predictive learning on graph-structured data (Kipf & Welling, 2017; Veličković et al., 2018; Hamilton et al., 2018; Xu et al., 2019; Corso et al., 2024). Despite their outstanding performance, unveiling the limitations of GNNs remains an active line of research. Notably, most works focus on the design of additional structural features of the graph to enhance the expressive power of the GNN (Srinivasan & Ribeiro, 2020; Souza et al., 2022; Zhang et al., 2023a; Wang et al., 2023; Graziani et al., 2024). Usually, these features add a non-negligible cost to the evaluation of the GNN and, similarly to the proposed LA-GFlowNets, there is a trade-off between computational complexity and expressivity (Morris et al., 2021). We refer the reader to the work of Papp & Wattenhofer (2022) for a comparison between GNN extensions in terms of their expressive power.

# B    MESSAGE-PASSING GNNS AND THE 1-WL TEST

Here, we denote a graph as a tuple $G = (V, E)$, where $V = \{1, 2, \dots, n\}$ is the set of nodes and $E \subseteq V \times V$ is the set of edges. More specifically, we consider attributed graphs — i.e., each node $v \in V$ has associated features $x_v \in \mathbb{R}^d$. Also, we denote the set of neighbors of a node $v$ in the graph as $\mathcal{N}_v = \{u : (u, v) \in E\}$.

**1-WL test.** The Weisfeiler-Lehman isomorphism test (Weisfeiler & Lehman, 1968) assigns colors for all nodes in an attributed input graph $G$ by applying the following iterative procedure:

*Initialization*: The colors of all nodes in $G$ are initialized using the initial node features: $\forall v \in V, c^0(v) = x_v$. If node features are not available, all nodes receive identical colors;

*Refinement*: At step $\ell$, the colors of all nodes are refined using a hash (injective) function: for all $v \in V$, we apply $c^{\ell+1}(v) = \text{HASH}(c^\ell(v), \{\!\{c^\ell(u) : (u, v) \in E\}\!\})$;

*Termination*: The test is carried out for two graphs in parallel and stops when the multisets of corresponding colors diverge, returning non-isomorphic. If the algorithm runs until the number of different colors stops increasing, the test is deemed inconclusive.

**Message-passing GNNs.**    Most popular GNNs can be described using the message-passing paradigm (Gilmer et al., 2017). Within this framework, a GNN initializes each node's embedding with its original features, i.e., $h_v^{(0)} = x_v$ for all node $v \in V$. Then, at each layer $\ell$, each node $v$ gathers messages from its neighbors $u \in \mathcal{N}_v$, compiling them into an aggregated message $m_v^{(\ell)}$:

$$m_v^{(\ell)} = \text{AGGREGATE}_\ell \left( \{\!\{h_u^{(\ell-1)} : u \in \mathcal{N}_v\}\!\} \right),$$

where AGGREGATE$_\ell$ is an arbitrary function on multisets, i.e., it is order-invariant. Subsequently, each node uses the (so-called) UPDATE function (e.g., a feedforward neural network) to refresh its embedding in light of the aggregated message:

$$h_v^{(\ell)} = \text{UPDATE}_\ell \left( m_v^{(\ell)}, h_v^{(\ell-1)} \right).$$

When both the AGGREGATE and UPDATE functions incur no loss of information (i.e., they are injective), the message-passing procedure coincides with the refinement step from the 1-WL test (Xu et al., 2019). In this case, the GNN attains the same power of the 1-WL isomorphism test. Conversely, the 1-WL serves as an upper-bound on the expressive power of message-passing GNNs.

## C   FORWARD-LOOKING- AND LED-GFLOWNETS

As mentioned earlier, both the forward-looking (FL-) and Learning Energy Decompositions (LED-) GFlowNets are built upon the principle of enhancing credit assignment via reparamaterizing the flow function $F$ as a logarithmic residual of a (either hand-crafted or learned) basic potential function $\phi$, i.e., $\log F(s, s') = \log \phi(s, s') + \log \tilde{F}(s, s')$, and $\tilde{F}(s, s') = \tilde{F}(s)p_F(s'|s)$ in the usual policy-based parameterization. Under this novel perspective, the DB loss becomes

$$\mathcal{L}_{DB}(p_F, p_B, F) = \mathbb{E}_\tau \left[ \frac{1}{\#\tau} \sum_{(s,s')\in\tau} \left( \phi(s,s') + \log \frac{\tilde{F}(s)p_F(s'|s)}{\tilde{F}(s')p_B(s|s')} \right)^2 \right], \tag{15}$$

with the constraint that $\sum_{(s,s')\in\tau} \phi(s,s') = \log R(x)$, in which $x$ is the unique terminal state in the trajectory $\tau$. Recall that, for FL-GFlowNets, $\phi(s,s') = \xi(s') - \xi(s)$ for a hand-crafted energy function $\xi$ such that $\xi(s_o) = 0$ and $\xi(x) = \log R(x)$ (Pan et al., 2023a, Assumption 1). For LED-GFlowNets, $\phi(s,s')$ is parameterized as an neural network taking as input a concatenation of the vectorial representations of $s$ and $s'$. The parameters of $\phi$ are then learned by minimizing

$$\mathcal{L}_{LS}(\tau) = \mathbb{E}_{(m_{s,s'})_{(s,s')\in\tau}\sim\text{Bernoulli}(1-\gamma)} \left[ \left( \frac{1}{\#\tau}\xi(x) - \frac{1}{C}\sum_{(s,s')\in\tau} m_{s,s'}\phi_\theta(s,s') \right)^2 \right], \tag{16}$$

in which $\{m_{s,s'}\}_{(s,s')}$ is a Dropout mask and $C = \sum_{(s,s')\in\tau} m_{s,s'}$ is the number of unmaked transitions. During training, we interleave gradient-based updates of the potential function $\phi$ and $(p_F, p_B, F)$ until a chosen stopping criterion (e.g., maximum number of epochs) is satisfied.

## D   PROOFS

This section contains self-contained and rigorous proofs for our theoretical results.

### D.1   PROOF OF REMARK 1

The terminal states of the modified flow network will have two types of nodes, with flow $\frac{F}{g^h}$ and $\frac{F}{g^h} + \delta_i$, with $\delta_i \geq 0$ and $\sum_{i=1}^{g^{h-1}} \delta_i = \delta$. We normalize those probabilities to obtain the individual probabilities for each terminal state, which determines the density of each sample. From that, we can proceed to compute the total variation distance between $\tilde{p}_T$ and $\pi$.

$$\|\tilde{p}_T - \pi\|_{TV} = \frac{1}{2}\sum_{x\in\mathcal{X}} |\tilde{p}_T(x) - \pi(x)|$$

$$= \frac{1}{2}\left[ (g^h - g^{h-1}) \left| \frac{F}{g^h}\frac{1}{F+\delta} - \frac{1}{g^h} \right| + \sum_{i=1}^{g^{h-1}} \left| \frac{F + g^h\delta_i}{g^h}\frac{1}{F+\delta} - \frac{1}{g^h} \right| \right]$$

$$= \frac{1}{2}\left[ \frac{g^h\delta - g^{h-1}\delta + \sum_{i=1}^{g^{h-1}}|g^h\delta_i - \delta|}{g^h(F+\delta)} \right].$$

We can lower bound $\sum_{i=1}^{g^{h-1}} |g^h\delta_i - \delta|$, by considering that $\sum_{i=1}^{g^{h-1}} (g^h\delta_i - \delta) = g^h\delta - g^{h-1}\delta$, taking the absolute value of the result and each element of the sum to obtain $g^h\delta - g^{h-1}\delta \leq \sum_{i=1}^{g^{h-1}} |g^h\delta_i - \delta|$.

Thus we obtain the lower bound

$$\frac{1}{2}\left[\frac{g^h\delta - g^{h-1}\delta + g^h\delta - g^{h-1}\delta}{g^h(F+\delta)}\right] \leq \frac{1}{2}\left[\frac{g^h\delta - g^{h-1}\delta + \sum_{i=1}^{g^{h-1}}|g^h\delta_i - \delta|}{g^h(F+\delta)}\right]$$

$$\left(1 - \frac{1}{g}\right)\frac{\delta}{F+\delta} \leq ||\tilde{p}_T - \pi||_{TV}.$$

This lower bound is reached when all error terms in the terminal states have the same value $\delta_i = \frac{\delta}{g^h}$.

To upper bound $|g^h\delta_i - \delta|$ we apply the triangle inequality, obtaining $|g^h\delta_i - \delta| \leq g^h\delta_i + \delta$ and $\sum_{i=1}^{g^{h-1}}|g^h\delta_i - \delta| \leq g^h\delta + g^{h-1}\delta$, from which we obtain the upper bound

$$||\tilde{p}_T - \pi||_{TV} \leq \frac{1}{2}\left[\frac{g^h\delta - g^{h-1}\delta + g^h\delta + g^{h-1}\delta}{g^h(F+\delta)}\right]$$

$$\leq \frac{\delta}{F+\delta}.$$

To obtain a tighter bound we break the sum $\sum_{i=1}^{g^{h-1}}|g^h\delta_i - \delta|$ by partitioning the sum into the first $I$ terms $S_A = g^h\sum_{i=1}^{I}|\delta_i - \frac{\delta}{g^h}|$ with $\delta_i < \frac{\delta}{g^h}$ and subsequent $g^{h-1} - I$ terms $S_B = g^h\sum_{j=I+1}^{g^{h-1}}|\delta_j - \frac{\delta}{g^h}|$ with $\delta_j \geq \frac{\delta}{g^h}$. By construction, we know that $S_A + g^h\sum_{i=1}^{I}\delta_i + g^h\sum_{j=I+1}^{g^{h-1}}\delta_j - S_B = g^{h-1}\delta$, simplifying to $S_B - S_A = \delta(g^h - g^{h-1})$. We rewrite $S_A + S_B = S_B - S_A + 2S_A = \delta(g^h - g^{h-1}) + 2S_A$, and by triangle inequality on $S_A$, we obtain the upper bound $\sum_{i=1}^{g^{h-1}}|g^h\delta_i - \delta| = S_A + S_B \leq g^h\delta - g^{h-1}\delta + 2I\delta$. Setting $I = g^{h-1} - 1$ (the biggest value it can have without breaking the constraints on $\delta_i$), it simplifies to $S_A + S_B \leq g^h\delta + g^{h-1}\delta - 2\delta$

$$||\tilde{p}_T - \pi||_{TV} \leq \frac{1}{2}\left[\frac{g^h\delta - g^{h-1}\delta + \sum_{i=1}^{g^{h-1}}|g^h\delta_i - \delta|}{g^h(F+\delta)}\right]$$

$$\leq \frac{1}{2}\left[\frac{g^h\delta - g^{h-1}\delta + g^h\delta + g^{h-1}\delta - 2\delta}{g^h(F+\delta)}\right]$$

$$\leq \left[\frac{g^h\delta - \delta}{g^h(F+\delta)}\right]$$

$$\leq \left(1 - \frac{1}{g^h}\right)\frac{\delta}{F+\delta}.$$

### D.2 PROOF OF THEOREM 1

To demonstrate this result, we will need the following facts regarding the function $f(x)\colon x \in \mathbb{R}^n \mapsto \sum_{i=1}^{n}|x_i - a_i|$ for positive constants $a_i$.

**Lemma 1** (Convexity). *Let* $\Delta_{n+1} = \{x \in \mathbb{R}^n\colon x_i \geq 0 \wedge \sum_{i=1}^{n}x_i = 1\}$ *and* $a \in \mathbb{R}^n$. *Then,* $f\colon \Delta_{n+1} \to \mathbb{R}$ *defined by* $f(x) = \sum_{i=1}^{n}|x_i - a_i|$ *is convex.*

*Proof.* It follows from $f(\alpha x + (1-\alpha)y) = \sum_{i=1}^{n}|\alpha x_i - \alpha a_i + (1-\alpha)y_i - (1-\alpha)a_i| \leq \alpha\sum_{i=1}^{n}|x_i - a_i| + (1-\alpha)\sum_{i=1}^{n}|y_i - a_i| = \alpha f(x) + (1-\alpha)f(y)$ for any $\alpha \in [0,1]$ and $x, y \in \Delta_{n+1}$. □

**Lemma 2** (Maximality at edges). *Let* $e_i \in \mathbb{R}^n$ *satisfy* $e_{ij} = 0$ *for* $j \neq i$ *and* $e_{ii} = 1$. *Then, the function* $f$ *from Lemma 1 achieves its maximum at* $\arg\max_{1 \leq i \leq n} f(e_i)$.

*Proof.* We will show that, for each $x \in \Delta_{n+1}$, there is a $i$ for which $f(e_i) \geq f(x)$. In particular, $f$ is maximized at one of the $e_i$'s. For this, note that

$$f(x) = f\left(\sum_{i=1}^{n}x_ie_i\right) \leq \sum_{i=1}^{n}x_if(e_i) \leq \max_{1 \leq i \leq n}f(e_i) \tag{17}$$

due to the convexity of $f$. Thus, $f$ is upper bounded by $\max_{1 \leq i \leq n} f(e_i)$. Conversely, there is a $e_i$ for which this upper bound is attained. Hence, $\arg\max_x f(x) \supseteq \arg\max_{1 \leq i \leq n} f(e_i)$. $\square$

**Lemma 3** (Minimality). *Let $f$ be the function of Lemma 1 and assume that $a \geq 0$ and $\sum_{i=1}^{n} a_i \leq 1$. Then, $f$ is minimized by $1 - \sum_{i=1}^{n} a_i$.*

*Proof.* Choose a $j \in \{1, \ldots, n\}$ arbitrarily. Since $x_j = 1 - \sum_{i=1, i \neq j}^{n} x_i$,

$$\sum_{i=1}^{n} |x_i - a_i| = \sum_{i=1, i \neq j}^{n} |a_i - x_i| + \left| a_j - 1 + \sum_{i=1, i \neq j}^{n} x_i \right| \geq \left| \sum_{i=1}^{n} a_i - 1 \right|. \tag{18}$$

Correspondingly, the lower bound in Equation 18 is achieved when $x_i = a_i$ for $i \neq j$ and $x_j = 1 - \sum_{i=1, i \neq j}^{n} a_i \geq 0$. This ensures that $f$ is minimized by $1 - \sum_{i=1}^{n} a_i$. $\square$

In words, Lemma 1 and Lemma 2 ensure that the TV distance between finitely supported distributions is convex and attains its maximum at a Dirac delta.

*Proof of Theorem 1.* Initially, let $\delta_x$ be the amount of extra flow reaching $x \in \mathcal{X}$ and define $\beta_x = \delta_x/\delta$. Then,

$$\|p_T - \tilde{\pi}\|_{TV} = \frac{1}{2} \sum_{x \in \mathcal{X}} |p_T(x) - \pi(x)| = \frac{1}{2} \sum_{x \in \mathcal{D}_{s^\star}} |p_T(x) - \pi(x)| + \frac{1}{2} \sum_{x \in \mathcal{D}_{s^\star}^c} |p_T(x) - \pi(x)|. \tag{19}$$

Since $p_T(x) = \tilde{\pi}(x)+\delta_x/F+\delta$ for $x \in \mathcal{D}_{s^\star}$ and $p_T(x) = \tilde{\pi}(x)/F+\delta$ for $x \in \mathcal{D}_{s^\star}^c$,

$$\sum_{x \in \mathcal{D}_{s^\star}^c} |p_T(x) - \pi(x)| = \frac{\delta}{F + \delta} \sum_{x \in \mathcal{D}_{s^\star}^c} \pi(x). \tag{20}$$

On the other hand,

$$\sum_{x \in \mathcal{D}_{s^\star}} |p_T(x) - \pi(x)| = \sum_{x \in \mathcal{D}_{s^\star}} \left| \frac{\tilde{\pi}(x) + \delta_x}{F + \delta} - \frac{\tilde{\pi}(x)}{F} \right| = \frac{\delta}{F + \delta} \sum_{x \in \mathcal{D}_{s^\star}} \left| \beta_x - \frac{\tilde{\pi}(x)}{F} \right|. \tag{21}$$

By Lemma 2, the function $f \colon \beta \mapsto \sum_{x \in \mathcal{D}_{s^\star}} |\beta_x - \pi(x)|$ is maximized at

$$\max_{y \in \mathcal{D}_{s^\star}} f(e_y) = \max_{y \in \mathcal{D}_{s^\star}} \sum_{x \in \mathcal{D}_{s^\star}} |e_{xy} - \pi(x)|$$

$$= \max_{y \in \mathcal{D}_{s^\star}} \left( \sum_{x \in \mathcal{D}_{s^\star}, x \neq y} \pi(x) \right) + (1 - \pi(y)) \tag{22}$$

$$= 1 + \sum_{x \in \mathcal{D}_{s^\star}} \pi(x) - 2 \min_{y \in \mathcal{D}_{s^\star}} \pi(y).$$

Similarly, Lemma 3 ensures that

$$\min_{\beta \in \Delta_{\#\mathcal{D}_{s^\star}+1}} f(\beta) = 1 - \sum_{x \in \mathcal{D}_{s^\star}} \pi(x). \tag{23}$$

Thus, since $\sum_{x \in \mathcal{D}_{s^\star}} \pi(x) = 1 - \sum_{x \in \mathcal{D}_{s^\star}^c} \pi(x)$,

$$\frac{\delta}{F + \delta} \left( 1 - \sum_{x \in \mathcal{D}_{s^\star}} \pi(x) \right) \leq \|p_T - \pi\|_{TV} \leq \frac{\delta}{F + \delta} \left( 1 - \min_{y \in \mathcal{D}_{s^\star}} \pi(y) \right). \tag{24}$$

$\square$

### D.3  PROOF OF THEOREM 2

As stepping stones towards proving Theorem 2, we first lay down Lemma 4 and Lemma 5.

**Lemma 4.** *Let $G = (V, E)$ and $G' = (V', E')$ be two non-isomorphic trees of size at most $n$. Let $\phi$ be the node embedding map of a 1-WL GNN with at least $2n - 1$ layers. Then, $\phi_v \neq \phi_{v'}$ for all $v \in V$ and $v' \in V'$.*

*Proof.* Recall 1-WL GNNs can distinguish any pair of non-isomorphic trees. Let $\mathcal{T}_n$ and $\mathcal{T}'_n$ denote the sets of computation trees (CTs) for each node in $G$ and $G'$ after $n$ layers, respectively. Likewise, let $\mathcal{T}_{2n-1}$ and $\mathcal{T}'_{2n-1}$ denote the sets of CTs after $2n+1$ layers. Since both graphs are non-isomorphic, 1-WL has already converged with $n$ steps — the maximum diameter of a tree is $n - 1$. Without loss of generality, $\mathcal{T}_n - \mathcal{T}'_n \neq \varnothing$, i.e., there is at least one CT in $\mathcal{T}_n$ that is not isomorphic to any tree in $\mathcal{T}'_n$. The same holds for $2n - 1$ layers, i.e., $\mathcal{T}_{2n-1} - \mathcal{T}'_{2n-1} \neq \varnothing$. Note that a CT $T_n \in \mathcal{T}_n - \mathcal{T}'_n$ is also a subtree of any $T_{2n-1} \in \mathcal{T}_{2n-1}$. Since $T_n \notin \mathcal{T}'_n$, $T_n$ is not a subtree of any CT in $\mathcal{T}'_{2n-1}$ — otherwise it would be in $\mathcal{T}'_n$ too. In other words, $\mathcal{T}_{2n-1} \cap \mathcal{T}'_{2n-1} = \varnothing$, implying directly our claim. □

**Lemma 5.** *Let $G = (V, E)$ and $G' = (V', E')$ be any two trees of size at most $n$, i.e., $|V|$ and $|V'| \leq n$. Also, let $I = (U, \emptyset)$ and $I' = (U', \emptyset)$ be graphs comprising isolated nodes, and $\phi$ be the node embedding map of a 1-WL GNN with at least $2n - 1$ layers. If $\{\phi_v, \phi_u\} = \{\phi_{v'}, \phi_{u'}\}$ for any $(v, u) \in V \times U$ and $(v', u') \in V' \times U'$, then the trees $(V \cup \{u\}, E \cup \{(v, u)\})$ and $(V' \cup \{u'\}, E' \cup \{(v', u')\})$ are isomorphic.*

*Proof.* If $\{\phi_v, \phi_u\} = \{\phi_{v'}, \phi_{u'}\}$, then we either have that *i)* $\phi_v = \phi_{v'}$ and $\phi_u = \phi_{u'}$ or *ii)* $\phi_v = \phi_{u'}$ and $\phi_{v'} = \phi_u$. In the first case, we can apply Lemma 4 to conclude that $G \cong G'$ (with associated bijection $g_1$). Since $\phi_u = \phi_{u'}$, we know that $x_u = x_{u'}$ and the corresponding singleton graphs are trivially isomorphic as well (with bijection $g_2$). Finally, we can build a bijection $g$ between the vertices of the merged graphs by making $g(v) = g_1(v)$ if $v \in V$ and $g(u) = g_2(u) = u'$. For the second case, Lemma 4 implies $G$ and $G'$ are singletons with $x_u = x_{v'}$ and $x_v = x_{u'}$. The result is a totally disconnected graph, except for an edge linking nodes with identical features in both graphs. □

Armed with the previous lemmata, Theorem 2 is straightforward assuming GNN depth $2n - 1$. From Lemma 5, we know that the action embeddings for any two nodes have an empty intersection. Likewise, two actions have the same embedding only if they leave from the same state and arrive at the same state. Therefore, all edges in the SG receive different embeddings. Recall that GNN embeddings are fed to MLP layers, which are universal approximators given enough width. Therefore, a 1-WL GNN followed by MLP can approximate any policy forward $p_F$. The same applies to the backward policy $p_B$. We can use the same combination to get state embeddings, which allow approximating any node flow function $F$. Therefore, we can choose the triplet $(p_F, p_B, F)$ respecting the DB conditions, for instance.

### D.4  PROOF OF THEOREM 3

Assume there is a 1-WL GFlowNet sampling from $\pi$. Since $\mathcal{G}$ is tree-structured, the mass arriving at $T(s_1) \cup T(s_2)$ must arrive through $s$ — i.e., all paths from $s_0$ to some $x \in T(s_1) \cup T(s_2)$ traverse $s$. Furthermore, there is no directed path from $s'$ to any terminal in $T(s'')$ or vice-versa, otherwise the skeleton (i.e., undirected structure) of $\mathcal{G}$ would contain a cycle. Then, $F(s, s') = \sum_{x \in T(s')} R(x)$ and $F(s, s') = \sum_{x \in T(s')} R(x)$, implying $F(s, s') \neq F(s, s'')$.

### D.5  PROOF OF THEOREM 4

Since child embeddings are included as additional inputs to LA-GFlowNets, it follows directly that LA-GFlowNets are at least as expressive as 1-WL GFlowNets. We are left with showing the converse does not hold. In Figure 5, we provide a construction for which 1-WL GFlowNets fail but LA-GFlowNets do not.

### D.6  PROOF OF THEOREM 5

Firstly, let $S = \{x_1, \ldots, x_B\} \subseteq \mathcal{X}$ and

$$e(S) = \frac{1}{2} \sum_{x \in S} \left| \frac{p_T(x)}{p_T(S)} - \frac{R(x)}{R(S)} \right|, \tag{25}$$

in which $p_T(S) = \sum_{x \in S} p_T(x)$ and $R(S) = \sum_{x \in S} R(x)$, as the TV distance between the restrictions of $p_T$ and $R$ to $S$. For conciseness, we write $p_T^{(S)}(x) = {}^{p_T(x)}/_{p_T(S)}$ and $\pi^{(S)}(x) = {}^{R(x)}/_{R(S)}$. We also denote by $\pi(x) = {}^{R(x)}/_{R(\mathcal{X})}$ the normalized reward in $\mathcal{X}$. Similarly, we define $e(p) = \mathbb{E}_{S \sim p}[e(S)]$. Then, we first show that $e(p) = 0$ when $\mathrm{TV}(p_T, \pi) = 0$. For this, note that $\mathrm{TV}(p_T, \pi) = 0$ implies $p_T(x) = \pi(x)$ for every $x$ and hence $p_T(S) = \pi(S) \; \forall S \subseteq \mathcal{X}$. Thus,

$$e(p) := \mathbb{E}_{S \sim p} \left[ \frac{1}{2} \sum_{x \in S} |p_T^{(S)}(x) - \pi^{(S)}(x)| \right] = 0. \tag{26}$$

On the other hand, assume that $e(p) = 0$. Recall that $p$ is a distribution of full support over $\{S \subseteq \mathcal{X} : |S| = B\}$ and that $B \geq 2$. In particular, $e(p)$ ensures that

$$e(S, \theta) := \frac{1}{2} \sum_{x \in S} \left| \frac{p_T(x)}{p_T(S)} - \frac{\pi(x)}{\pi(S)} \right| = 0. \tag{27}$$

Hence, $p_T(S)\pi(x) = \pi(S)p_T(x)$ for each $S$ and $x \in S$. Write then $S = S' \cup \{x\}$ and conclude that $p_T(S')\pi(x) = \pi(S')p_T(x)$ for every $S'$ and $x \notin S'$. Thus, by summing both members of this equality across $x' \notin S'$, we notice that

$$p_T(S')(1 - \pi(S')) = \pi(S')(1 - p_T(S')), \tag{28}$$

i.e., $p_T(S') = \pi(S')$. Thus, by iterating this procedure, we conclude that $p_T(x) = \pi(x)$ for all $S'$ and $x \notin S'$. Since $S'$ and $x$ were chosen arbitrarily, $p_T(x) = \pi(x)$ for every $x \in \mathcal{X}$. Consequently, $\mathrm{TV}(p_T, \pi) = 0$. This ensures the equivalence between $e(p)$ and $\mathrm{TV}(p_T, \pi)$ in terms of characterizing the GFlowNet's distributional correctness.

### D.7  PROOF OF COROLLARY 1

Recall the definition of $e(S)$ in Equation 25. We start demonstrating that $P_S(\cdot; \beta)$ is indeed a probability distribution. Clearly, $P_T(S; \beta) \geq 0$ for every $S \subseteq \mathcal{X}$. On the other hand,

$$\sum_{S \subseteq \mathcal{X}} P_S(S; \beta) = \sum_{S \subseteq \mathcal{X}, \#S = \beta} \binom{n-1}{\beta-1}^{-1} \underbrace{\sum_{x \in S} p_T(x)}_{p_T(S)}$$

$$= \sum_{S \subseteq \mathcal{X}, \#S = \beta} \binom{n-1}{\beta-1}^{-1} p_T(S) \tag{29}$$

$$= \sum_{x \in \mathcal{X}} \binom{n-1}{\beta-1}^{-1} \binom{n-1}{\beta-1} p_T(x) = 1,$$

since each $p_T(x)$ appears exactly $\binom{n-1}{\beta-1}$ times on the sum above. Hence, $P_S(\cdot; \beta)$ is a probability distribution. As for the rest, let $\hat{e} = \mathbb{E}_{S \sim p}[e(S)]$, $\#\mathcal{X} = n$, $\mathcal{P}_\beta = \{S \subseteq \mathcal{X} : \#S = \beta\}$, and $\Delta = \frac{n}{2\beta} \max_{S \in \mathcal{P}_\beta} |p_T(S) - \pi(S)|$. We will first show that

$$\mathrm{TV}(p_T, \pi) - \hat{e} \leq \Delta. \tag{30}$$

Then, we will verify that $\mathrm{TV}(p_T, \pi) - \hat{e} \geq -\Delta$. These inequalities will jointly imply Corollary 1. In this scenario, note there are $\binom{n-1}{\beta-1}$ subsets of $\mathcal{X}$ with $\beta$ elements containing a $x \in \mathcal{X}$. Thus,

$$\mathrm{TV}(p_T, \pi) = \frac{1}{2} \sum_{S \in \mathcal{P}_\beta} \sum_{x \in S} \binom{n-1}{\beta-1}^{-1} |p_T(x) - \pi(x)|. \tag{31}$$

For conciseness, define $d_{TV} = TV(p_T, \pi)$. Hence,

$$
\begin{aligned}
d_{TV} - \hat{e} &= \frac{1}{2} \sum_{S \in \mathcal{P}_\beta} \sum_{x \in S} \binom{n-1}{\beta-1}^{-1} |p_T(x) - \pi(x)| - P_S(S) \left| \frac{p_T(x)}{p_T(S)} - \frac{\pi(x)}{\pi(S)} \right| \\
&\leq \frac{1}{2} \sum_{S \in \mathcal{P}_\beta} \sum_{x \in S} \binom{n-1}{\beta-1}^{-1} \left( \left| p_T(x) - \frac{p_T(S)}{\pi(S)} \pi(x) \right| + \pi(x) \left| 1 - \frac{p_T(S)}{\pi(S)} \right| \right) \\
&\qquad\qquad\qquad\qquad\qquad\qquad - \frac{P_S(S)}{p_T(S)} \left| p_T(x) - \frac{\pi(S)}{p_T(S)} \pi(x) \right| \\
&= \frac{1}{2} \sum_{S \in \mathcal{P}_\beta} \sum_{x \in S} \binom{n-1}{\beta-1}^{-1} \pi(x) \left| 1 - \frac{p_T(S)}{\pi(S)} \right| \\
&= \frac{1}{2} \binom{n-1}{\beta-1}^{-1} \sum_{S \in \mathcal{P}_\beta} |p_T(S) - \pi(S)| \\
&\leq \frac{1}{2} \binom{n-1}{\beta-1}^{-1} \binom{n}{\beta} \max_{S \in \mathcal{P}_\beta} |p_T(S) - \pi(S)| = \frac{n}{2\beta} \Delta
\end{aligned}
\tag{32}
$$

since $P_S(S)/p_T(S) = \binom{n-1}{\beta-1}^{-1}$ and there are $\binom{n}{\beta}$ $\beta$-sized subsets of $\mathcal{X}$. For the reverse inequality, notice that

$$
\begin{aligned}
d_{TV} - \hat{e} &= \frac{1}{2} \sum_{S \in \mathcal{P}_\beta} \sum_{x \in S} \binom{n-1}{\beta-1}^{-1} |p_T(x) - \pi(x)| - P_S(S) \left| \frac{p_T(x)}{p_T(S)} - \frac{\pi(x)}{\pi(S)} \right| \\
&\geq \frac{1}{2} \sum_{S \in \mathcal{P}_\beta} \sum_{x \in S} \binom{n-1}{\beta-1}^{-1} |p_T(x) - \pi(x)| \\
&\qquad\qquad - P_S(S) \left( \left| \frac{p_T(x)}{p_T(S)} - \frac{\pi(x)}{p_T(S)} \right| + \left| \frac{\pi(x)}{p_T(S)} - \frac{\pi(x)}{\pi(S)} \right| \right) \\
&= -\frac{1}{2} \sum_{S \in \mathcal{P}_\beta} \binom{n-1}{\beta-1}^{-1} p_T(S) \sum_{x \in S} \pi(x) \left| \frac{1}{p_T(S)} - \frac{1}{\pi(S)} \right| \\
&= -\frac{1}{2} \binom{n-1}{\beta-1}^{-1} \sum_{S \in \mathcal{P}_\beta} |p_T(S) - \pi(S)| \geq -\frac{n}{2\beta} \max_{S \in \mathcal{P}_\beta} |p_T(S) - \pi(S)| .
\end{aligned}
\tag{33}
$$

### D.8 PROOF OF COROLLARY 2

Again, recall the definition of $e(S)$ in Equation 25. We now provide a self-contained proof of Corollary 2, which follows from Corollary 1 and Hoeffding's inequality Alquier (2024). Firstly, let $\hat{e} = \mathbb{E}_{S \sim p}[e(S)]$ and $e_i = e(S_i)$. Since $\hat{e} - e_i \in [-1, 1]$, Hoeffding's inequality yields

$$
\mathbb{E} \left[ \exp \left\{ \lambda \left( \hat{e} - \frac{1}{m} \sum_{1 \leq i \leq m} e_i \right) \right\} \right] \leq \exp \left\{ \frac{\lambda^2}{2m} \right\}.
\tag{34}
$$

Then, Chernoff's bound implies

$$
\mathbb{P}_{S_1, \ldots, S_m} \left[ \hat{e} \geq \frac{1}{m} \sum_{1 \leq i \leq m} e_i + s \right] \leq \mathbb{E} \left[ \exp \left\{ \lambda \left( \hat{e} - \frac{1}{m} \sum_{1 \leq i \leq m} e_i \right) \right\} \right] e^{-\lambda s} \leq \exp \left\{ \frac{\lambda^2}{2m} - \lambda s \right\}
$$

due to Equation 34. This upper bound is minimized when $\lambda = sm$. In this case, $\lambda^2/2m - \lambda s = -s^2 m/2$. By letting $s = -2 \log \delta / m$, we verify that

$$
\mathbb{P}_{S_1, \ldots, S_m} \left[ \hat{e} \geq \frac{1}{m} \sum_{1 \leq i \leq m} e_i + \sqrt{\frac{2 \log \frac{1}{\delta}}{m}} \right] \leq \delta.
\tag{35}
$$

Then, Corollary 1 and the complementary of the preceding inequality imply

$$\mathbb{P}_{S_1,\dots,S_m}\left[\mathrm{TV}(p_T,\pi) \leq \frac{1}{m}\sum_{1\leq i\leq m} e_i + \max_{S\subseteq\mathcal{X},|S|=B}|p_T(S) - \pi(S)| + \sqrt{\frac{2\log\frac{1}{\delta}}{m}}\right] \geq 1 - \delta. \quad (36)$$

### D.9 PROOF OF PROPOSITION 1

As detailed Appendix C, the global minimizer of both FL- and LED-GFlowNets' learning objectives satisfy $\sum_{(s,s')\in\tau}\phi(s,s') = -\log R(x)$ for every trajectory $\tau$. Since $\mathcal{L}_{\mathrm{LED}}(s,s') = 0$,

$$\tilde{F}(s)\exp\{\phi_\theta(s,s')\}p_F(s,s') = p_B(s',s)\tilde{F}(s')$$

for every trajectory finishing at $x$. Therefore, for every trajectory $\tau \rightsquigarrow x$,

$$p_F(\tau) = p_B(\tau|x)\frac{\tilde{F}(x)}{\tilde{F}(s_o)}\prod_{(s,s')\in\tau}\exp\{-\phi(s,s')\}$$

$$= p_B(\tau|x)\frac{\tilde{F}(x)}{\tilde{F}(s_o)}\exp\left\{-\sum_{(s,s')\in\tau}\phi(s,s')\right\}$$

$$= p_B(\tau|x)\frac{\tilde{F}(x)}{\tilde{F}(s_o)}R(x).$$

Hence,

$$p_T(x) = \sum_{\tau\rightsquigarrow x}p_F(\tau) = \sum_{\tau\rightsquigarrow x}\frac{\tilde{F}(x)R(x)}{\tilde{F}(s_o)}p_B(\tau|x) \propto \tilde{F}(x)R(x)\sum_{\tau\rightsquigarrow x}p_B(\tau|x) = \tilde{F}(x)R(x), \quad (37)$$

ensuring that the marginal distribution learned by terminally unrestricted FL- and LED-GFlowNets does not necessarily match GFlowNet's target distribution.

# E  EXPERIMENTAL DETAILS

We provide further details regarding the experimental setup for each section below. Experiments were run in a cluster equipped with A100 GPUs, using a single GPU per run. We also include a more extensive discussion of existing empirical results and additional experiments in Section E.1 and Section E.2, respectively. Section E.3 describes our implementation of both FL- and LED-GFlowNets and their terminally unrestricted variants. All experiments relied on Adam (Kingma & Ba, 2014) with a learning rate of $10^{-3}$ for $p_F$ and $10^{-2}$ for $\log Z$ for stochastic optimization (Madan et al., 2022).

## E.1  EXPERIMENTS FOR SECTION 3

**Set generation.**  The support $\mathcal{X}$ is defined as the collection of sets with 16 elements sampled from a deposit $\mathcal{D} = \{1, \ldots, 32\}$. To define the reward function, we let $f \colon \mathcal{D} \to \mathbb{R}$ with $f(d) \sim \mathcal{U}[0, 1]$ and let $\log R(x) = \sum_{d \in x} f(d)$. We implemented an MLP with 2 256-dimensional hidden layers to parameterize both the forward policy and the flow function. For the weighting function $\gamma$, we note that $\#\mathcal{D}_{s'} = \binom{32 - |s'|}{16 - |s'|}$, in which $|s'|$ is the current state's size.

**Sequence design.**  The support $\mathcal{X}$ is defined as the collection of sequences of size up to 12 with elements extracted from a deposit $\mathcal{D} = \{1, \ldots, 4\}$. We implemented an MLP with 2 256-dimensional hidden layers for both the forward policy and flow functions, both of which received as input a sequence of length 12 padded with 0s. Then, the reward function of a $\mathbf{x} \in \mathbb{R}^8$ is defined by $f \colon \mathcal{D} \to \mathbb{R}$ and $g \colon [[1, 12]] \to \mathbb{R}$,, with $f(d), g(i) \sim U[-1, 1]$ for $d \in \mathcal{D}$ and $i \in [[1, 12]]$, through $\log R(x) = \sum_i f(x_i) g(x_i)$. For the weighting function $\gamma$, we note that $\#\mathcal{D}_{s'} = 1 + 4 + \cdots + 4^{12 - |s'|}$ is the number of $s'$'s terminal descendants.

**Phylogenetic inference.**  A *phylogenetic tree* is defined by a complete binary tree $\mathcal{G}$ with labeled leaves corresponding to observed biological species and anonymous internal nodes corresponding to their evolutionary ancestors. Also, we consider a set $\mathbf{Y} \in \mathbb{R}^{32 \times 7}$ of DNA sequences of size 32 associated to the 7 observed species; the likelihood of $\mathbf{Y}$ is defined by J&C69 (Jukes & Cantor, 1969)'s mutation model and computed by Felsenstein's algorithm (Felsenstein, 1981), and the reward function is the unnormalized posterior induced by an uniform prior distribution over trees. We adopt the iterative process proposed by Zhou et al. (2024) to sample phylogenetic trees with GFlowNets, and use a Graph Isomorphism Network (Xu et al., 2019) to parameterize $p_F$. For the weighting function $\gamma$, we recall that $\#\mathcal{D}_{s'} = (2 \cdot (7 - |s'|) - 1)!!$ is the number of terminal descendants of $s'$, with $|s'|$ as the amount of connected components in $s'$ (?).

**Hypergrid.**  The support $\mathcal{X}$ is composed of the tuples $(x, y) \in [0, 11] \times [0, 11]$ characterizing the $12 \times 12$ 2-dimensional grid. The generative process is identical to that of Bengio et al. (2021); Malkin et al. (2022; 2023). The reward function, in particular, is defined by

$$R(\mathbf{s}) = 10^{-3} + 0.5 \prod_{1 \le i \le 2} \mathbf{1}_{\{|s_i/11| \in (0.25, 0.5]\}} + 2 \prod_{1 \le i \le 2} \mathbf{1}_{\{|s_i/11| \in (0.3, 0.4)\}} \tag{38}$$

for a state $\mathbf{s} = (s_1, s_2)$; $\mathbf{1}_A$ is the indicator function of the event $A$. Similarly to Madan et al. (2022), we use a batch size of 16 trajectories and train the model for 62500 epochs ($10^6$ trajectories). We parameterize the forward policy with a MLP composed of 2 128-dimensional layers. Also, $\#\mathcal{D}_{s'} = (11 - s_1')(11 - s_2') + 1$ is the number of terminal states reachable from a state $s'$.

**Details on the experiments for Figure 3.**  To further understand Theorem 1 in the context of the training GFlowNets, we show in Figure 3 the average log-squared balance violation along trajectories for the generative tasks considered in Section 3. As expected, the magnitude of the DB loss is mostly dominated by early-transitions of the generative process. Also, this dominance is more notorious for the problems of set generation and phylogenetic inference and less noticeable for the problems of sequence design and hypergrid navigation, consistently with the results observed in Figure 4 concerning the improved performance of minimizing our weighted loss in Equation 6 with respect to the traditional approach. In this regard, we emphasize that the design of sequences and hypergrid navigation are the only tasks in Figures 3 and 4 with variably sized trajectories. When training a model for these tasks, we observe that most sampled trajectories are relatively short in the initial stages of training, which may hamper the improvements enacted by our weighted learning objective due to the similarities of the weights. Nonetheless, as we acknowledge in Section 6, a deeper understanding of the influence of each transition to the accuracy of the GFlowNet's sampling distribution is still required. In any case, it is clear that the conventional uniform weighting in Equation 6 is sub-optimal. Importantly, we also believe that the best choice for $\gamma$ should be considered in a problem-by-problem basis.

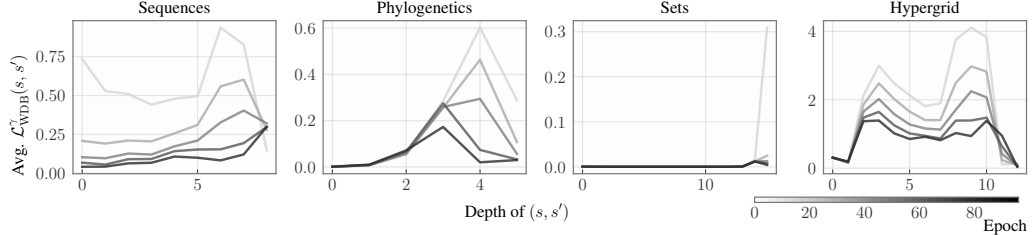

Figure 11: **Average** $\mathcal{L}_{\text{WDB}}^{\gamma}(s, s') := \gamma(s, s') \left( \log(F(s)p_F(s, s') - \log(F(s')p_B(s, s'))) \right)^2$ **on random trajectories** during training. Results are averaged across 5 runs. In contrast to Figure 3, we observe that $\gamma$ has a *skewing-then-smoothing effect*, i.e., it initially assigns large weights to terminal states (where the training signal, $R$, is received), then equalizes the weights across transitions. This is noticeable for the Phylogenetics and Set tasks, which are the ones in which $\mathcal{L}_{\text{WDB}}^{\gamma}$ performs the best.

### E.2 EXPERIMENTS FOR SECTION 4

**Setup for Figure 6.** This experiment is built upon simple 3-state SGs with the form $L \leftarrow P \rightarrow R$, in which $P$ is a 3-regular graph of 8 nodes and $L$ and $R$ are $P$'s non-isomorphic children obtained by the addition of a single edge. In particular, we choose four different tuples $(L_i, P_i, R_i)$ for the four plots of Figure 6. See Figure 9 for an illustration of 2 of the implemented SGs. To parameterize the policies of both LA- and the standard GFlowNets, we use a 3-layer

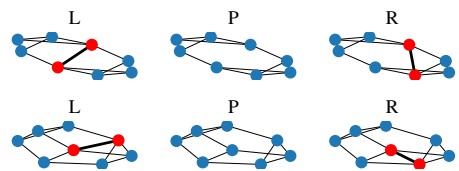

Figure 9: Examples of tuples $(L_i, P_i, R_i)$. Added edges and their nodes are highlighted.

GIN (Xu et al., 2019) having 32-dimensional layer, followed by an MLP of 2 32-dimensional layers. For LA-GFlowNet, the MLP's input size is twice as large as the one for the standard model.

**Runtime analysis for LA-GFlowNets.** From a theoretical standpoint, the time-complexity of a LA-GFlowNet grows linearly with the maximum number $C$ of children of a state in the state graph. In contrast, the cost of a conventional GFlowNet implementation is constant with respect to $C$. Please see Figure 10 for a comparison between the training times of LA-GFlowNet, a standard GFlowNet (trained by minimizing TB), and a GFlowNet trained via the flow matching objective (FM) — see (Bengio et al., 2021, Equation 11) — for the set generation task with the same hyperparameters described in Section E.1. We report the average running time for 16 epochs averaged across 5 seeds. Investigating whether an equivalent boost in expressivity can be achieved at a lower complexity is a promising future direction.

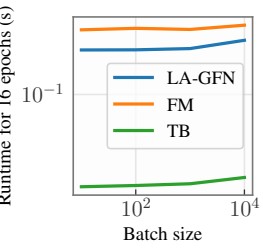

Figure 10: Training times.

### E.3 EXPERIMENTS FOR SECTION 5

**FL- and LED-GFlowNets.** We followed the experimental setup of Pan et al. (2023a) and Jang et al. (2024) when implementing both FL- and LED-GFlowNets. To avoid implementation bias, we reproduced our experiments using Pan et al. (2023a)'s publicly released code[1] and obtained similar results. In particular, both $p_F$ and $\phi$ were parameterized with MLPs. For LED-GFlowNet, we carried out 8 stochastic gradient steps for learning $\phi$ for each epoch during training. For the standard GFlowNet trained by minimizing the TB loss, we followed Malkin et al. (2022)'s instructions.

We privately exchanged emails with the main author of LED-GFlowNets (Jang et al., 2024) regarding whether their implementations enforced $F(x) = R(x)$ or not. Importantly, he told us he verified their experiments and that their code for the set environment indeed did not enforce $F(x) = R(x)$, thereby implementing the terminally unrestricted GFlowNet variant described in Proposition 1, but explained to us his remaining experiments were correct. He also told us that the base code for his experiments on the set environment was borrowed from Pan et al. (2023a)'s work.

---

[1]Available online at github.com/ling-pan/FL-GFN.

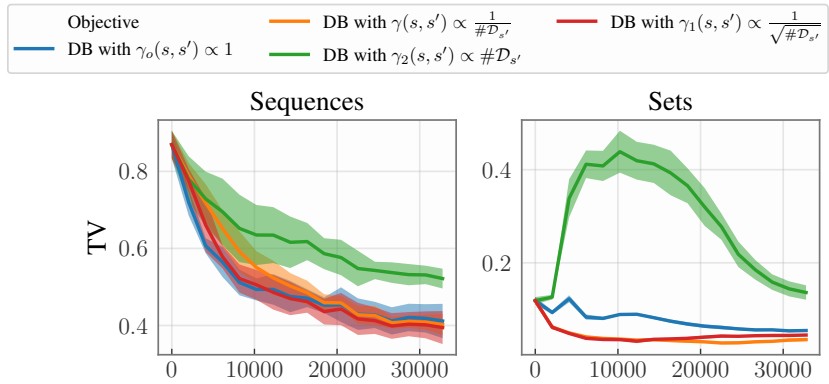

Number of observed trajectories

Figure 12: **Illustration of the effect of $\gamma$ on the learning convergence of GFlowNet.** We introduce two novel weighting functions: $\gamma_1$ and $\gamma_2$. On the one hand, $\gamma_1$ weights each transition in inverse proportion to the square root of its number of descendants $\#\mathcal{D}_{s'}$ and, as expected, produces a behavior similar to that of the originally proposed $\gamma$. On the other hand, $\gamma_2$ — which is the inverse of $\gamma$ — significantly hampers the learning convergence of the trained GFlowNet.

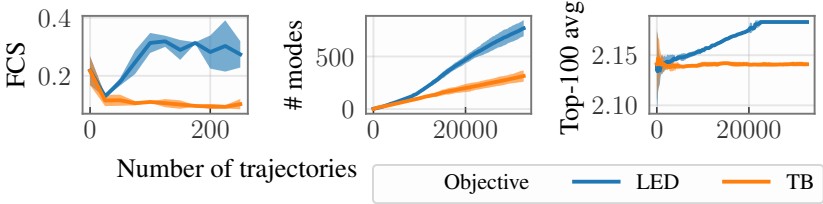

Number of trajectories

Figure 13: **Complement to Figure 8 for the PHO4 task** (Shen et al., 2023, Section 7). Again, FCS is the only metric that properly reflects the GFlowNet's distributional accuracy. Indeed, despite quickly covering the high-probability regions of the target (mid and right panels), TU-LED-GFlowNet fails to learn the right distribution (left panel). Results are averaged over three runs.

**Set generation.** The experimental setup is identical to the one described at Section 3. To compute Shen et al. (2023)'s accuracy, we pre-computed the average of $R(x)$ under the target distribution by extensively enumerating the SG's terminal states. For FCS, we randomly sampled 32 batches of terminal states of size up to 128 for the Monte Carlo estimator.

**Bag generation.** The experimental setup is mostly the same we used for set generation. However, due to the space of bags being significantly larger than the space of sets, we fix $\mathcal{D} = \{1, \ldots, 16\}$ and generate 8-sized multisets with elements in $\mathcal{D}$.

### E.4 ADDITIONAL EXPERIMENTS

**Transition-wise losses for $\mathcal{L}^{\gamma}_{\text{WDB}}$.** Figure 11 shows the averages of $\mathcal{L}^{\gamma}_{\text{WDB}}(s, s')$ as a function of the transition $(s, s')$'s depth along randomly sampled trajectories. We note that, while the standard DB loss is mostly dominated by earlier states on the initial training stages (see Figure 3), $\mathcal{L}^{\gamma}_{\text{WDB}}$ is governed by near-terminal transitions. Hence, the primary training signal, which corresponds to the reward of terminal states, receives a larger weight during the optimization process. On the other hand, as training progresses, the loss variability within a trajectory decreases, guiding the GFlowNet to learn a balanced flow across the entire state graph — essential for accurate distributional approximation (Theorem 1).

**Additional weighting schemes for $\mathcal{L}^{\gamma}_{\text{WDB}}$.** In the light of Figure 3, Theorem 1, and of the above analysis, we deduce that an effective weighting function $\gamma$ should prioritize near-terminal transitions. Figure 11 validates this intuition on the tasks of set generation and sequence design (see Figure 4) by

showing that, when $\gamma$ is a strictly increasing function of the transition's depth, the resulting $\mathcal{L}_{\text{WDB}}^{\gamma}$ performs competitively with or better than the standard DB loss. In practice, these results can guide the design of an appropriate weighting function. However, a principled approach for optimally constructing $\gamma$ remains open, being an important venue for future investigations.

**Effectiveness of FCS in a real dataset.** We reproduce the experiments in Figure 8 for the task of sampling DNA sequences of length 10 in proportion to a reward function defined by wet-lab measurements of the sequence's binding affinity to a yeast transcription factor (PHO4) (Shen et al., 2023; Jain et al., 2022; Barrera et al., 2016; Trabucco et al., 2022). For this, we follow the experimental setup described in Section E.3 for the sequence design task. Consistently with Figure 8, Figure 13 and Table 2 show that FCS is the only tractable metric able to correctly infer the provable incorrectness of TU-LED-GFlowNet. Indeed, in terms of mode discovery and Shen et al. (2023)'s accuracy, TU-LED-GFlowNet outperforms the TB-GFlowNet — despite being farther from the target distribution. We omit FL-GFlowNet since there is no natural candidate for the potential function.

