# OpenReview forum: "When do GFlowNets learn the right distribution?"
_ICLR.cc/2025/Conference — ICLR 2025 Spotlight_

### Official Review · Reviewer_MUz7 · 2024-11-03

**Soundness:** 3
**Presentation:** 3
**Contribution:** 3
**Rating:** 8
**Confidence:** 2

**Summary:**

This paper investigates the theoretical foundations of Generative Flow Networks for distributions over discrete and compositional objects. The paper evaluates the impact of violations in the detailed balance of the underlying flow network on the correctness of GFlowNet's sampling distribution. They demonstrate that the effect of imbalanced edges is influenced by the total amount of flow passing through them. The paper also explores the representational limits of GNN-based GFlowNets, and shows that they cannot correctly sample from certain state graphs and target distributions. To address these limitations, the authors propose the new method, LA-GFlowNets, and a metric to evaluate GFlowNets.

**Strengths:**

1. The paper tries to understand theoretical foundations of the popular GFlowNets, which is important and insightful. It produces theoretical guarantees and evaluate it both theoretically and empirically.

2. The paper clearly shows the contributions in Table 1, which makes the paper easier to follow.

3. The experiments are organized and easy to follow. The evaluation process is logical and comprehensive.

**Weaknesses:**

Two tiny points:

1. In Theorem 4, the paper says "LA-GFlowNet is more powerful". How to define "powerful" here?

2. I think the notations in Background is a little complicated. It might be better to use a figure to illustrate the model.

**Questions:**

In Figure 7, why does FL-GFlowNets have such large variance compared to other methods?

---

> ### Author Response · Authors · 2024-11-19
>
> Thank you for reviewing and acknowledging the importance of our work. We will include the discussion below into the updated manuscript to improve the clarity of our work.
>
> ## Weaknesses
>
> > 1. In Theorem 4, the paper says "LA-GFlowNet is more powerful". How to define "powerful" here?
>
> Thanks for the opportunity to clarify our arguments. In a nutshell, we use the terms “powerful” and “expressive” interchangeably: a family of neural networks is considered more powerful than another if it can realize a broader set of functionals.
>
> By stating that LA-GFlowNets are strictly more powerful than standard GFlowNets, we mean that every flow assignment problem solvable by a GFlowNet can also be solved by a LA-GFlowNet —  and that the converse is not true. This is the central idea in Theorem 4. In this regard, we also presented in Figure 5 a collection of distributions from which a LA-GFlowNet can sample, but a standard GFlowNet cannot.
>
> For coherence, we replaced the word “powerful” with “expressive” in Theorem 4.
>
> > 2. I think the notations in Background is a little complicated. It might be better to use a figure to illustrate the model.
>
> We have added a figure in the background section illustrating the concepts of state graph, forward, and backward policies. We hope that this enhances the readability of our work. Otherwise, we are welcome to additional suggestions.
>
>
> ## Questions
>
> > 1.  In Figure 7, why does FL-GFlowNets have such large variance compared to other methods?
>
> Thank you for the question. The main reason for the large variance of FL- and (to a lesser but significant extent) LED-GFlowNets in Figure 7’s top-100 reward panel (now Figure 8) is the absence of a unique minimizer of their respective learning objectives (as stated in Proposition 1). Consequently, modifying the initialization of the neural network that parameterizes the policy network can lead to potentially drastic changes in the model’s equilibrium distribution and in the rate with which the high-probability regions of the target are discovered. We also note that a similarly high variance was observed in Figure 2 of FL-GFlowNet’s original work by Pan et al. (2023).
>
> Pan et al. Better Training of GFlowNets with Local Credit and Incomplete Credit Assignment. ICML 2023.
>
> We appreciate your interest and suggestions to improve our work. We will be glad to discuss the answers above in more detail, should any clarifications be needed.

---

> > ### Comment · Reviewer_MUz7 · 2024-11-25
> > **Thanks for the response**
> >
> > Thank author for their response. After reviewing it, I decide to keep supporting this paper.

---

### Official Review · Reviewer_2KxY · 2024-11-03

**Soundness:** 4
**Presentation:** 3
**Contribution:** 4
**Rating:** 8
**Confidence:** 4

**Summary:**

Generative Flow Networks (GFlowNets) are a recent class of deep generative models suitable for representing probability distributions over discrete and compositional data structures. The paper asks a temporary and practically important question, which, to the best of my knowledge, has not been answered until now: Do GFlowNets correctly learn the target distribution? The paper answers this question in three parts.

The first part investigates how a small perturbation of the detailed balance condition affects the total variation (TV) distance between the sampling and target distributions. The authors manifest their analysis by the average of the summands in the detailed balance loss for different depths of the state transitions. Nicely, this theoretical and empirical finding is then put to practical use by designing a weighted detailed balance loss, which assigns the state transitions with different weights. Consequently, the authors achieve better (or on par) performance than other state-of-the-art training objectives.

The second part focuses on the parameterization of GFlowNets as the cause of violating the detailed balance. The authors select the popular domain where the target distribution is a distribution over graphs, where the parameterization of GFlowNets is thus instantiated with graph neural networks, and are concerned with the standard questions about the one-hop Weisfeiler-Leman (WL) test and its impact on the representational power of GFlowNets. The authors show that a 1-WL GFlowNet can approximate any target distribution over trees but not graphs with two 1-WL indistinguishable nodes. This analysis leads to the design of the look-ahead GFlowNets, which offer more representational power than the 1-WL GFlowNet. This synthesis is again confirmed by experimental evidence.

The third part is concerned with assessing the goodness-of-fit of GFlowNets in cases where the learned distribution is intractable. The authors propose Flow Consistency in Subgraphs (FCS) as the expected total variation distance between $\beta$-sized restrictions of the marginal distribution of the forward policy and the reward function. Moreover, the authors discuss the equivalence between FCS and TV distance and provide PAC statistical guarantees. The empirical results demonstrate that the FCS metric is a computationally efficient and close approximation of the TV distance.

**Strengths:**

The paper is very dense and comprehensive.

Most of the theoretical findings are nicely supported by experimental evidence.

The writing is excellent. Although some parts would deserve minor improvements (more on that below).

**Weaknesses:**

Minor:

The writing is excellent, as mentioned above. The whole paper works extensively with the sampling distribution and the target distribution. However, these need to be adequately defined. I would expect to see a more precise statement of these two distributions, e.g., in the first paragraph of Section 2. For instance, in line 143, the authors start using ``*GFlowNets sampling distribution*'' without showing the symbol for it. Right after that, the authors say that their target is $R$, and then, only in line 185, we see that $\pi\propto R$. The first mention that $\pi$ is the target is in line 202. The first appearance of `$\pi$ is defined to be equal to ...' can be seen in line 398.

Line 120:  ``*Finally, a flow is a function ...*'' It should be a function $\mathcal{T}\rightarrow\mathbb{R}_{+}$, where $\mathcal{T}\subset\mathcal{S}$ is a set of complete trajectories (Definition 7 in Bengio et al. 2023).

Line 123: *``a SG''* -> *``an SG''*

Line 127: *``an uniform''* -> *``a uniform''*

Line 200: Which tasks in Section 2 do the authors have in mind? Do the authors mean those mentioned in the last paragraph of Section 3? The authors first refer to Figure 2 in Section 3.1, but the four tasks are first introduced in Section 3.2. The sequence of ideas can be improved.

Figure 4: *``state graph''* -> *``state graphs''* It would be more readable to separate the two state graphs. Their blending is confusing.

*``i.d.d.''* is inconsistent with *``wrt''* throughout the text.

Lines 458 and 462: *``a FL-''* -> *``an FL-''*

Line 459: *``part''* -> *``a part''*

**Questions:**

Line 370: ``*For most benchmark tasks, e.g., hypergrid environment (Malkin et al., 2023), set generation (Shen et al., 2023), and sequence design (Jain et al., 2022), we can exactly and efficiently compute $p_T$.*'' Please clarify if the statement about efficiently computing $p_T$ for benchmark tasks means you can compute arbitrary marginal distributions of $p_T$ over subsets of $x$. If so, could you provide more details on how this is done efficiently?

For Figure 2, please clarify if the results are averaged over multiple runs. If so, specify how many runs were performed and what aspects (e.g., initialization) varied between runs.

Could it be possible to show an equivalent of Figure 2 for Avg. $\mathcal{L}^{\gamma}_{\text{WDB}}$ to see a comparison to the standard DB loss? This could help illustrate the impact of the weighting scheme on different transitions.

What is the shaded area for each curve in Figure 3? Is it the standard deviation or the interquartile range? How was it computed?

---

> ### Author Response · Authors · 2024-11-19
>
> Thank you for thoughtfully reading and appreciating our work. We will adopt all of your suggestions to improve the text. Also, we provide further clarifications for specific issues below. We will adjust the text accordingly.
>
> ## Weaknesses
>
> > The whole paper works extensively with the sampling distribution and the target distribution
>
> We value your suggestions. To improve the clarity of the manuscript, we have included an explicit definition of the GFlowNet’s sampling $p_{T}$ and target $\pi$ distributions on Line 132 (Section 2).
>
> > Line 120: ``Finally, a flow is a function ...''
>
> Thank you for the opportunity to clarify this. This is a notational convenience that was (maybe unfortunately) canonized in the GFlowNet literature; in Bengio et al. (2021), a flow function F simultaneously represents the flow through a *trajectory* (Definition 6, $F \colon \mathcal{T} \rightarrow \mathbb{R}$), through a *set of trajectories* (Definition 6, $F \colon 2^{\mathcal{T}} \rightarrow \mathbb{R}$), and through a *state* (Definition 7, $F \colon \mathcal{S} \rightarrow \mathbb{R}$). In our work, we utilize the latter definition. We will make this explicit in the updated manuscript.
>
> > Line 200: Which tasks in Section 2 do the authors have in mind? Do the authors mean those mentioned in the last paragraph of Section 3?
>
> We thank the reviewer for catching this typo. We have rewritten this sentence to emphasize that the results are empirically validated in Section 3.2 (instead of Section 2) and illustrated in Figure 2 (now Figure 3).
>
> “We experimentally validate these findings for common benchmark tasks in Section 3.2 (see Figure 3).“
>
> > Figure 4: "state graph'' -> "state graphs'' It would be more readable to separate the two state graphs. Their blending is confusing.
>
> In truth, Figure 4 (now Figure 5) represents a single state graph. To make this clearer, we replaced “A pair of state graph and reward function” with “A combination of state graph and reward function” in the figure’s caption.
>
> ## Questions
>
> > … If so, could you provide more details on how this is done efficiently?
>
> We are happy to discuss this further. For autoregressive problems (such as biological sequence design (Jain et al. 2022)), there is a single trajectory $\tau$ leading to each terminal state $x$. As a consequence, the marginal of $x$ can be readily computed as $p_{T}(x) = p_{F}(\tau)$. For the set generation and hypergrid navigation tasks, $p_{T}$ can only be efficiently computed when the corresponding state graphs are relatively small. Under these conditions, we can exhaustively enumerate the trajectories leading to a fixed state $x$ and exactly compute the summation in Equation (8). We have included this discussion in the main text.
>
> Jain et al. Biological Sequence Design with GFlowNets. ICML 2022.
>
> To avoid potential confusion, we will replace the word “efficiently” with “tractably”.
>
> > For Figure 2, please clarify if the results are averaged over multiple runs.
>
> Thank you for pointing this out — the results in Figure 2 (now Figure 3) were indeed representative of a single run. For completeness, we executed additional experiments with five different random seeds and computed the expected loss along a trajectory averaged across the runs. As we can see in the updated figure, our conclusions regarding the non-uniform impact of the transition-wise terms in the DB loss are preserved.
>
> > Could it be possible to show an equivalent of Figure 2 for Avg. $\mathcal{L}_{\mathrm{WDB}}^{\gamma}$ to see a comparison to the standard DB loss?
>
> Thank you for the suggestion. We have included a counterpart of Figure 2 (now Figure 3) for the $\mathcal{L}_{\mathrm{WDB}}^{\gamma}$ loss in Figure 11 on page 25 of the submitted PDF. For the tasks of phylogenetic inference and set generation, our weighting scheme clearly assigns a large value to the initially imbalanced terminal transitions — where the reward function, which is the sole training signal, is evaluated. Throughout training, this heterogeneity is gradually reduced towards uniformization. For the remaining problems (hypergrid navigation and sequence design), the effect of $\gamma$ on the transition-wise distribution of the loss appears to be negligible. Importantly, these observations are consistent with our results in Figure 4. Overall, these results show that our weighting scheme — although not harmful — is open to improvements. This is an important research line inaugurated by our work.
>
> > What is the shaded area for each curve in Figure 3?
>
> The shaded area represents a one-standard-deviation distance from the average TV of three randomly initialized runs. This detail will be emphasized in the updated manuscript. We are grateful for the reviewer’s thorough attention to our work.
>
> Thanks again for the attentive feedback. If our corrections do not satisfactorily address your concerns, please let us know. We will be more than happy to discuss these matters in further detail.

---

> > ### Comment · Reviewer_2KxY · 2024-11-25
> >
> > I want to thank the authors for a detailed reply to my comments. Their answers are excellent. I will keep a positive assessment of the paper.

---

### Official Review · Reviewer_ykTe · 2024-11-03

**Soundness:** 3
**Presentation:** 2
**Contribution:** 3
**Rating:** 6
**Confidence:** 3

**Summary:**

The paper provides a detailed analysis of Generative Flow Networks (GFlowNets) to understand when and whether they accurately learn target distributions. The authors extend the DB loss by non-uniformly weighting the transition-wise terms to account for this heterogeneity. For graph-structured generation tasks, they introduce LA-GFlowNets to boost the expressive power of GNN-based GFlowNets by incorporating the embeddings of the children of a state into the policy network. The paper also introduces Flow Consistency in Subgraphs (FCS), a new metric for assessing GFlowNet performance, arguing that it provides a more reliable measure than popular protocols.

**Strengths:**

1. The question is well-defined. The authors start with analysis and findings, and then propose their method and new evaluation metrics.
2. As for the paper presentation, key contributions are well-articulated, with a clear breakdown of findings and their implications for GFlowNets.
3.. This paper proposes a new metric FCS, which has the potential to standardize GFlowNet evaluation, addressing a critical gap in current methodologies. And the paper provides a detailed comparison of FCS with traditional evaluation metrics.
4. The paper includes theoretical formulation and proofs as well as empirical results to validate their findings.

**Weaknesses:**

1. The legends, captions and ticks in figures are too small, which is not very readable for readers.
2. Some expressions are not well-written or formal. e.g. 'Figure 7 and Table 2 teach us three facts.'
3. Some figures lack legends (e.g. Figure 5). Though the authors use different colors in captions to distinguish, it's may still lead to confusion.
4. The weighted detailed balance (WDB) loss design (e.g. the choice of $\gamma$ ) appears heuristic without detailed discussion on optimizing these weights or examining the variance across tasks, which needs more detailed explanation.
5. The experiments predominantly focus on controlled scenarios, limiting insight into real-world application suitability.

**Questions:**

1. Could the authors elaborate on the choice of the $\gamma$ for the WDB loss? Are there alternative weighting functions that could improve performance in specific applications?
2. As for other applications such as NLP and molecule generation, could the authors provide more insights on whether FCS metric can be generalizable?

---

> ### Author Response · Authors · 2024-11-19
> **Official Comment by Authors (1/2)**
>
> Thank you for carefully reviewing our work and for the suggestions to improve its readability. We address each of your concerns below, and have incorporated all your suggestions.
>
>
> ## Weaknesses
>
>
> > 1. The legends, captions and ticks in figures are too small, which is not very readable for readers.
>
>
> Thank you for the suggestion. We have increased the size of the captions and ticks of the figures in the revised PDF that we have uploaded. We hope that this enhances their readability.
>
>
> > 2. Some expressions are not well-written or formal. e.g. 'Figure 7 and Table 2 teach us three facts.'
>
>
> We have changed the sentence to “There are three main takeaways from Figure 7 and Table 2.” (now Figure 8). We are open to additional feedback.
>
>
> > 3. Some figures lack legends (e.g. Figure 5). Though the authors use different colors in captions to distinguish, it's may still lead to confusion.
>
>
> Thanks for pointing this out — indeed, caption-only labeling might be confusing. Based on your feedback, we have now added legends to Figure 6 (formerly Figure 6).
>
>
> >  4. The weighted detailed balance (WDB) loss design (e.g. the choice of $\gamma$) appears heuristic without detailed discussion on optimizing these weights or examining the variance across tasks, which needs more detailed explanation.
>
>
> That’s another excellent suggestion.
>
>
> To further illustrate the impact of $\gamma$ on the training of GFlowNets, we have included additional experiments with two different choices for $\gamma$: $\gamma\_{1}(s, s') \propto \frac{1}{\sqrt{\\# \mathcal{D}\_{s'}}}$ and $\gamma\_{2}(s, s’) \propto \\# \mathcal{D}\_{s’}$. On the one hand, $\gamma_{2}$ prioritizes the transitions near terminal states in the state graph, similarly to the original $\gamma$. On the other hand, $\gamma_{1}$ assigns larger weights to transitions near the initial state (it is the inverse to the $\gamma$ in the main text). As expected, Figure 12 on page 25 of the updated manuscript shows that both $\gamma_{1}$ and the original $\gamma$ result in similar improvements to the learning convergence of the GFlowNet; $\gamma_{2}$, in contrast, significantly hinders the model’s training efficiency.
>
>
>
>
> In this context, there are two primary conclusions to be drawn from these experiments. First, the conventional uniform weighting scheme is suboptimal. Second, an effective $\gamma$ should be an increasing function of the transition’s depth.
> Thank you for the opportunity to shed light on these aspects.  We hope our detailed explanation and these additional insights foster further research toward the optimal design of $\gamma.
>
>
> > 5. The experiments predominantly focus on controlled scenarios, limiting insight into real-world application suitability.
>
>
> Thank you for pointing this out. To underscore the promise of our approach in real-world settings, we have included experiments on DNA sequence design with wet-lab measurements of binding affinity to a yeast transcription factor (PHO4) as a reward. Please refer to Shen et al. (2023, Section 7) for further details on these experiments. As it is mostly unclear how to devise a FL-like reparameterization for these tasks, we exclude it from our results.
>
>
> In this context, Figure 13 and Table 3 on page 26 of the revised PDF confirm that FCS is uniquely capable of assessing the distributional accuracy of GFlowNets. More specifically, in contrast to FCS, both the mode-discovery rate as well as the approach by Shen et al. (2023) assign a high score to a distributionally incorrect terminally unconstrained GFlowNet.
>
>
> We hope that this additional experiment provides further insight on the applicability of FCS on real-world tasks. Regarding the metric’s effectiveness in problems such as NLP and drug discovery, please look at our answer to Question#2 below.
>
>
> Shen et al. Towards Understanding and Improving GFlowNet Training. ICML 2023.

---

> ### Author Response · Authors · 2024-11-19
> **Official Comment by Authors (2/2)**
>
> ## Questions
>
>
> > 1. Could the authors elaborate on the choice of the $\gamma$ for the WDB loss? Are there alternative weighting functions that could improve performance in specific applications?
>
> We hope that our analysis in Weakness#4 above provided a clearer understanding of the role of $\gamma$ in the WDB objective. As we noted, different choices of $\gamma$ might lead to similar improvements in learning convergence.
>
> Moreover, our discussion highlights the significance of choosing $\gamma$ wisely. In this context, one particularly interesting direction is exploring whether $\gamma$ can be effectively learned, thereby tailoring it to the needs of specific applications.
>
> >  2. As for other applications such as NLP and molecule generation, could the authors provide more insights on whether FCS metric can be generalizable?
>
> Thank you for the question. From a practical perspective,  FCS is the best computationally tractable metric for assessing the accuracy of a learned GFlowNet. Based on the results from Section 5, we expect FCS to be robust against false positives even in tasks such as NLP and molecule generation — in contrast to the traditional evaluation protocols presented therein. As suggested by Corollary 2, it is unlikely that a small FCS and a large TV distance are simultaneously observed. In a broader context, we believe that FCS has the potential to become the go-to standard for benchmarking GFlowNets.
>
> Thank you for your constructive and thoughtful feedback that has helped to strengthen this work. We hope our answers and additional empirical evidence have satisfactorily addressed your concerns, and would be grateful if the same could be reflected in your stronger support for this work.

---

> > ### Comment · Reviewer_ykTe · 2024-11-20
> >
> > I thank the authors for their responses. I will maintain the positive score.

---

### Author Response · Authors · 2024-11-19

Dear reviewers and AC,

Thank you for your service.

We are pleased that reviewers found our work well-written (ykTe, 2KxY, MUz7), theoretically well-grounded (ykTe, MUz7), empirically well-supported (2KxY, MUz7), and potentially impactful (ykTe).

We also thank the reviewers for their thoughtful feedback and contributions to strengthen the clarity of our work. Additional experiments are included in Section E.4 at the end of the updated manuscript. Further explanations and corrected typos are highlighted in yellowish-brown for improved readability.

Best regards,

Authors.

---

### Author Response · Authors · 2024-12-01

Dear reviewers and chairs,

We are grateful for the reviewers' support for our work and commitment to the peer reviewing process. All experiments, corrections, and discussions were incorporated into the revised manuscript.

Best regards,

Authors.

---

### Meta-Review · Area_Chair_aRHu · 2024-12-26

**Metareview:**

The paper offers a comprehensive analysis of Generative Flow Networks (GFlowNets), focusing on their ability to accurately learn target distributions. The paper addresses an important and timely question in GFlowNets, with strong theoretical analysis and practical contributions. The introduction of FCS provides a promising new approach to standardizing GFlowNet performance evaluation. The theoretical findings are well-supported by experiments, and the writing is generally clear and organized. The consensus was in favor of accepting the paper.

**Additional Comments On Reviewer Discussion:**

there are some issues with presentation, including small text and unclear legends in figures, which hinder readability. Additionally, some expressions are informal or unclear, reducing the overall clarity. The design of the weighted DB loss is not fully explained, especially regarding its optimization or variations across tasks. Finally, the experiments primarily focus on controlled settings, which limits their relevance to real-world applications. It would be great the authors incorporate these concerns is revising the paper

---

### Decision · Program_Chairs · 2025-01-22

Accept (Spotlight)